organic chemistry/crystallography/biochemistry

monoamine oxidase inhibitors, antidepressant, quinazoline analogues, structure–activity relationships

**Authors for correspondence:**
Adel Amer
e-mail: aaamer@taibahu.edu.sa; adel.amer@alex-sci.edu.eg
Rona R. Ramsay
e-mail: rrr@st-andrews.ac.uk

Dedication: Dedicated to the memory of Prof. Hans W. Zimmer (1921–2001).

This article has been edited by the Royal Society of Chemistry, including the commissioning, peer review process and editorial aspects up to the point of acceptance.

# Design, synthesis, molecular modelling and *in vitro* screening of monoamine oxidase inhibitory activities of novel quinazolyl hydrazine derivatives

Adel Amer[1,2], Abdelrahman H. Hegazi[2], Mohammed Khalil Alshekh[1], Hany E. A. Ahmed[3,4], Saied M. Soliman[2], Antonin Maniquet[5] and Rona R. Ramsay[5]

[1]Department of Chemistry, College of Science, Taibah University, Al-Madinah Al-Munawarah, Saudi Arabia
[2]Department of Chemistry, Faculty of Science, Alexandria University, Alexandria 21321, Egypt
[3]Pharmacognosy and Pharmaceutical Chemistry Department, Pharmacy College, Taibah University, Al-Madinha Al-Munawarah, Saudi Arabia
[4]Pharmaceutical Organic Chemistry Department, Faculty of Pharmacy, Al-Azhar University, Cairo, Egypt
[5]Biomedical Sciences Research Complex, University of St Andrews, Biomolecular Sciences Building, North Haugh, St Andrews KY16 9ST, UK

AA, 0000-0003-0481-475X; AHH, 0000-0003-1364-676X;
SMS, 0000-0001-8405-8370; RRR, 0000-0003-1535-4904

A new series of N'-substituted benzylidene-2-(4-oxo-2-phenyl-1,4-dihydroquinazolin-3(2H)-yl)acetohydrazide (**5a–5h**) has been synthesized, characterized by FT-IR, NMR spectroscopy and mass spectrometry and tested against human monoamine oxidase (MAO) A and B. Only (4-hydroxy-3-methoxybenzylidene) substituted compounds gave submicromolar inhibition of MAO-A and MAO-B. Changing the phenyl substituent to methyl on the unsaturated quinazoline ring (**12a–12d**) decreased inhibition, but a less flexible linker (**14a–14d**) resulted in selective micromolar inhibition of hMAO-B providing insight for ongoing design.

# 1. Introduction

Depression has been reported to be the fourth global burden of disease, with nearly 12% of the global disability-adjusted life years [1]. In the Kingdom of Saudi Arabia, improvements in socioeconomic status have been shown to be associated with increased chronic diseases including chronic mental diseases like depression [2–7]. Monoamine oxidases (MAO), enzymes containing covalently bound flavin adenine dinucleotide (FAD) as the cofactor, are located on the mitochondrial outer membrane where they oxidize various physiologically and pathologically important monoamines. These neurotransmitters and hormones include serotonin, noradrenaline and dopamine that function to regulate movement, emotion, reward, cognition, memory and learning [8–10].

The two isoforms of MAO (MAO-A and MAO-B) are characterized by different affinities for inhibitors and different specificities for substrates [11,12]. MAO-A preferentially metabolizes serotonin, adrenaline, and noradrenaline [13], whereas 2-phenylethylamine and benzylamine are predominantly metabolized by MAO-B [14]. Tyramine and dopamine are common substrates for both isoenzymes [15]. The therapeutic interest in monoamine inhibitors (MAOIs) covers two major categories: MAO-A inhibitors are used mostly in the treatment of mental disorders, in particular depression and anxiety [16–18], whereas MAO-B inhibitors are used in the treatment of Parkinson's disease and are of interest against Alzheimer's disease [19,20].

A major goal of our group is to develop compounds capable of inhibiting monoamine oxidase A as a target for the treatment of depression and psychological disorders. We have designed and synthesized several series of compounds carrying 3-benzylquinoxaline and quinazoline scaffolds with selective activity towards MAO-A [21–25]. Based on the results, we decided to change the scaffold from benzyl quinoxaline to methyl and phenyl quinazolines as the prominent motif for discovery of variety of biologically active compounds. To further explore the structure–activity relationships of the new quinazoline class of MAO-A inhibitors, we have now synthesized and evaluated a series of N'-substituted benzylidene-2-(6-chloro-4-oxo-2-aryl-1,4-dihydroquinazolin-3(2H)-yl)acetohydrazides and N'-substituted benzylidene-2-(2-methyl-4-oxoquinazolin-3(4H)-yl)acetohydrazides for their ability to inhibit the enzymatic activity of both MAO isoforms. The rationale of work design was built upon different factors that include: (i) the presence of a hydrophobic nitrogen heterocyclic head (Quinazolyl moiety), (ii) the presence of the hydrazido functionality (–CO–NH–N=) connected to the N atom via a saturated carbon linker, and (iii) the presence of an electron-rich aromatic tail ($Y = OH$, OMe in type **5** compounds) (chart 1). We also prepared and examined a series of (*E*)-3-(substituted benzylideneamino)-2-methylquinazolin-4(3H)-ones which contain imino functionality (–HC=N–) linker (compounds **14**).

# 2. Results and discussion

## 2.1. Organic synthesis

The synthetic route for the target compounds **5a–5h** is depicted in scheme 1. Allowing isatoic anhydride **1a–1b** to react with methyl glycine hydrochloride and triethylamine afforded the intermediate methyl (2-aminobenzoyl)glycinate **2a–2b**. The cyclocondensation of compound **2a–2b** and aromatic aldehyde in ethanol under reflux processed to furnish **3a–3d**. One-pot three components synthetic route to type **3** was achieved using metal oxides nanoparticles [26]. Allowing isatoic anhydride to react with methyl glycine hydrochloride, triethylamine and aromatic aldehyde in ethanol–water mixture in the presence of catalytic amount of copper oxide nanoparticles led to the formation of **3a–3d** in good yields. The treatment of the ester **3a–3d** with hydrazine hydrate afforded the corresponding hydrazides **4a–4d**. Their structures were confirmed by spectroscopic tools and secured by X-ray single crystal analysis of compound **4c** as a prototype (figure 1; electronic supplementary material, table S1).

Reaction of compounds **4a–4d** with different aromatic aldehydes in ethanolic solution and keeping the reaction mixture warm and stirring overnight furnished the desired products N'-benzylidene-2-(4-oxo-2-phenyl-1,4-dihydroquinazolin-3(2H)yl)acetohydrazide derivatives **5a–5h**. The NMR spectra of type **5** compounds reveal that they exist in two isomeric forms *E* and *Z* in dimethyl sulfoxide solution [27,28]. As a prototype, [1]H NMR spectrum of **5d** (figure 2) showed two sets of two doublets in 64 : 36 ratio for the α-CH$_2$ with coupling constant of 17.6 and 16.4 Hz at $\delta$ 3.64, 4.03 and 3.34, 4.53, respectively. The hydrazono N=CH appeared as singlets at $\delta$ 7.82 and 8.03. In [13]C NMR, the α-C resonated at $\delta$ 45.42 and 46.32. These assignments were supported by the [1]H–[1]H COSY NMR spectrum which confirmed

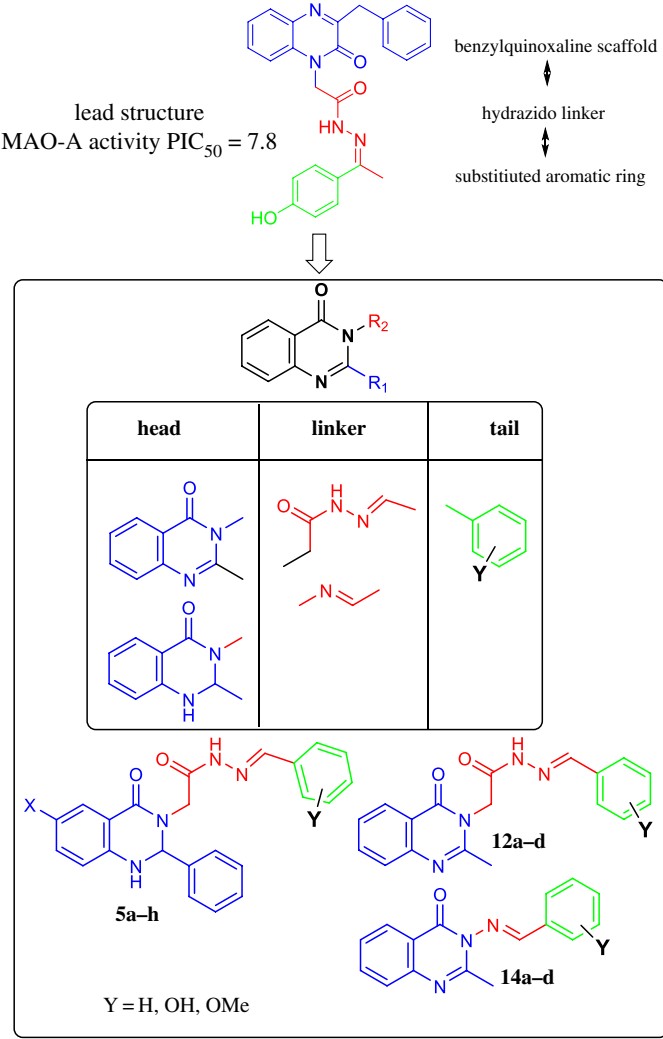

**Chart 1.** Planned modification and newly designed quinazolyl hydrazine derivative monoamine inhibitors.

the correlation assignments of α-CH$_2$ and provided support to full peak assignments. The $sp^3$CH resonated at δ 6.00 and correlated to two signals at δ 7.25 and 7.33 ppm of the NHs.

On the other hand, an attempt to accelerate the reaction **4b** and vanillin using few drops of acetic acid and heating the reaction mixture to reflux led to ring opening and the precipitation of **6**, as evident by NMR. The methylene protons showed one signal at δ 3.64 that correlates to carbon at δ 44.75 in heteronuclear single quantum coherence spectroscopy (HSQC) spectrum indicating the loss of the $sp^3$CH chiral centre.

To synthesize the quinazolin-3(4H)-yl)acetohydrazide analogue **7** (scheme 2), the methyl 2-(4-oxo-2-phenyl-1,4-dihydroquinazolin-3(2H)-yl)acetate **3a** was oxidized with 2,3-dichloro-5,6-dicyano-1,4-benzoquinone (DDQ) to afford methyl 2-(4-oxo-2-phenylquinazolin-3(4H)-yl)acetate **8** [29]. Hydrazinolysis of **8** with hydrazine hydrate furnished **7** which was confirmed by spectral data and its melting point matches the literature [30].

For the comparative study, compounds **12a–12d** were prepared using a method modified from the one reported earlier [31,32] (scheme 3) starting from 2-methyl-4H-benzo[d][1,3]oxazin-4-one **9** to **10** which upon hydrazinolysis furnished **11**. The reaction of compound **11** with aromatic aldehydes in dimethylformamide and few drops of acetic acid furnished N′-substituted benzylidene-2-(2-methyl-4-oxoquinazolin-3(4H)-yl)acetohydrazide **12a–12d** in good yield. According to the obtained NMR spectra, these compounds exist as a mixture of E and Z (see Experimental section) and not as reported earlier as a single isomers [31,32].

The reaction sequence that led to the synthesis of (E)-3-(substituted benzylideneamino)-2-methylquinazolin-4(3H)-one **14a–14d** is shown in scheme 4. Thus, compound **9** was converted to **13** by reacting with hydrazine hydrate [33], which upon condensation with various aromatic aldehydes

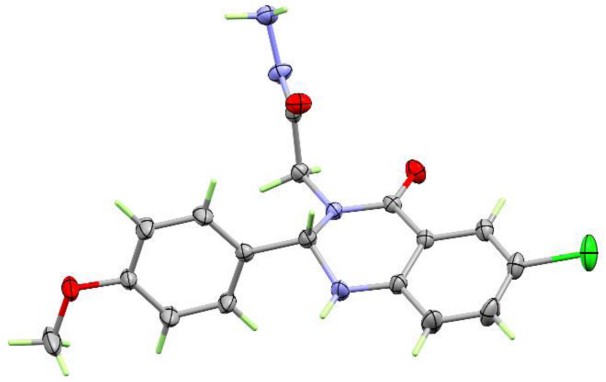

**Scheme 1.** Synthesis of compounds **3**, **4**, **5** and **6**.

**Figure 1.** Perspective drawing of compound **4c**.

**Figure 2.** Structures of the *E/Z* isomers of **5d**.

**Scheme 2.** Synthetic route to compound **7**.

a   Ar=C$_6$H$_5$                  E/Z 80 : 20
b   Ar=4-OHC$_6$H$_4$              E/Z 77 : 23
c   Ar=4-OMeC$_6$H$_4$             E/Z 78 : 22
d   Ar=3-OMe, 4OHC$_6$H$_3$        E/Z 81 : 19

**Scheme 3.** Synthesis of compounds **12**.

a   Ar = C$_6$H$_5$
b   Ar = 4-OHC$_6$H$_4$
c   Ar = 4-OMeC$_6$H$_4$
d   Ar = 3-OMe, 4OHC$_6$H$_3$

**Scheme 4.** Synthetic route to compounds **14**.

**Table 1.** Inhibition of MAO activity.

| compound | % inhibition at 10 μM[a] | | IC$_{50}$ (μM)[b] | |
| | hMAO-A | hMAO-B | hMAO-A | hMAO-B |
|---|---|---|---|---|
| **3a** | −6 | 38 | none | 5.28 ± 1.44 (40%) |
| **3b** | 10 | 34 | | |
| **3c** | −3 | 10 | | |
| **3d** | 0 | 27 | | |
| **4b** | 6 | 31 | | |
| **4c** | 0 | 27 | | |
| **4d** | 0 | 28 | | |
| **5a** | 19 | 55 | 6.2 ± 0.95 | 10.0 ± 1.1 |
| **5b** | 6 | 36 | none | 83 |
| **5d** | 71 | 70 | 0.27 ± 0.06 | 0.75 ± 0.18 |
| **5e** | 10 | 19 | | |
| **5g** | 22 | 36 | 4.34 ± 1.31 | 11.0 ± 4.0 |
| **5h** | 70 | 72 | 0.31 ± 0.11 | 0.44 ± 0.1 |
| **6** | none | 12 | (rate increases)[c] | 9.0 ± 2.3 |
| **11** | 15 | 19 | | >100 |
| **13** | −8 | 23 | | |
| **12a** | 6 | 26 | | |
| **12b** | 16 | 16 | | >100 |
| **12c** | 5 | 21 | | |
| **12d** | 27 | 52 | 1.41 ± 0.47 (32%) | 3.62 ± 0.69 |
| **14a** | 5 | 40 | | 6.84 ± 0.76 (77%) |
| **14b** | 14 | 52 | | 4.64 ± 0.41 (77%) |
| **14c** | 8 | 44 | | 6.75 ± 1.06 (73%) |
| **14d** | 11 | 44 | | 5.93 ± 1.02 (77%) |
| safinamide | 11 | 90 | | 0.08 [39] |
| harmine | 100 | 17 | 0.009 ± 0.001 (100%) | 11.6 ± 3.1 |

[a]Observed inhibition (% = 100 × rate with inhibitor/rate without inhibitor). Values are average of two independent experiments with less than 15% difference.

[b]IC$_{50}$ values ± s.d. for two independent experiments with 22 points each. The maximum inhibition (%) is given in parentheses if less than 95%.

[c]This compound (**6**) is oxidized by MAO-A so is a substrate ($K_M$ = 35.5 μM) inducing H$_2$O$_2$ production in the absence of tyramine.

furnished **14a–14d** [34–38]. Their structures were found in agreement with the assigned molecular structure and confirmed by IR, NMR spectroscopy and mass spectrometry.

## 2.2. MAO inhibition

All compounds were screened for inhibition of the activity of membrane-bound human MAO-A and MAO-B at 10 μM in the presence of tyramine at $2 \times K_M$ (0.8 mM for MAO-A and 0.32 mM for MAO-B) (table 1). Apart from **5d**, **5h**, **11** and **12b**, which gave equal inhibition of MAO-A and MAO-B in this screen, the compounds were slightly selective for MAO-B (see electronic supplementary material, figure S1). For the compounds that gave more than 30% inhibition, the IC$_{50}$ was determined using seven concentrations of each compound (0.01–300 μM) with tyramine substrate at $2 \times K_M$. The MAO-A and MAO-B IC$_{50}$ values for these selected compounds are given in table 1 accompanied by the maximum extent of the inhibition observed. Incomplete inhibition of activity could arise either from

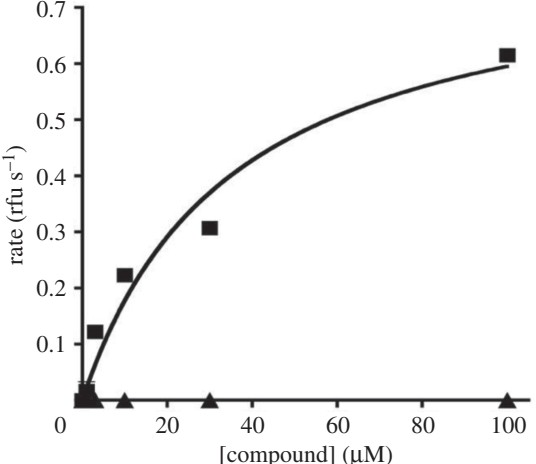

**Figure 3.** Oxidation of **6** (squares) but not **12d** (triangles) by MAO-A. The concentration of compound was varied in the standard assay coupled to the production of $H_2O_2$ but in the absence of the substrate tyramine.

indirect effects on these enzymes or from the compounds acting as substrates at the highest concentrations. Only compounds **5a–5h**, **12d** and **14a–14d** can be said to be classical inhibitors with **5a, 5h** and **12d** inhibiting both MAO-A and MAO-B. By contrast, **14a–14d** were highly MAO-B selective. The best inhibitors of MAO-A were **5d and 5h** with submicromolar $IC_{50}$ values ($0.27 \pm 0.06$ and $0.31 \pm 0.06\,\mu M$, respectively). These compounds were slightly less effective against MAO-B activity ($IC_{50}$ values 0.75 and $0.44\,\mu M$, respectively). Having both 3-$OCH_3$ and 4-OH substituents on the tail aryl group was important for high affinity, as seen also in **5h** and **12d**.

There was no difference in the inhibition without or with 30 min preincubation of MAO with the compounds. This and the restoration of full activity upon dilution ($\times 100$) indicated that the inhibition was reversible. For both MAO-A (with **5h** or **12d**) and MAO-B (with **12d** or **14b**), increasing the substrate concentration in the assay increased the rate of reaction, suggesting competitive inhibition.

Compound **6** in the screening assay with $10\,\mu M$ inhibitor gave an increased rate of appearance of fluorescence indicating faster production of the product $H_2O_2$ (not the slower rate observed with other compounds that inhibit MAO activity). When incubated with MAO-A in the absence of the normal substrate, **6** (but not **12d**) gave a measurable rate of $H_2O_2$ production, as shown in figure 3. Thus, **6** binds to and is oxidized by MAO-A. The $V_{max}$ for oxidation of **6** by MAO-A was 77% of the $V_{max}$ for tyramine under the same conditions and the $K_M$ was $35.5 \pm 7.8\,\mu M$.

## 2.3. Molecular docking simulation

The energy minimum structures of human MAO-A and MAO-B after removing covalent ligands (clorgyline for 2BXR and deprenyl for 2BYB) were used as the models (electronic supplementary material, figure S2).

The MAO-A binding site formed of a single cavity extends from the flavin ring to the cavity-shaping loop consisting of residues 210–216. The volume of this cavity is estimated to be approximately 550 Å$^3$ and is lined by aliphatic and aromatic residues, which are quite hydrophobic. In addition, two cysteine residues are located near the entry of the catalytic site that share in the interaction. The MAO-A active site lacks the MAO-B constriction allowing bulkier molecules to approach the aromatic cage of FAD, Tyr407 and Tyr444 [40].

The active site of hMAO-B is small in volume compared to MAO-A but shows the same hydrophobic nature. The MAO-B active site entrance is surrounded by hydrophobic amino acid residues (Leu171, Tyr326, Phe168, Ile198 and Ile199), followed by a narrow hydrophobic tunnel leading to the aromatic cage lined by the isoalloxazine of FAD, Tyr398 and Tyr435 [40].

The interactions of the phenylquinazoline head, acetohydrazide linker and aromatic tail with amino acid residues in the active sites of MAO-A and MAO-B are shown in figure 4 and listed in electronic supplementary material, table S1. The selective **14b** (E-form) derivative is well tolerated in the MAO-B pocket, binds close to the FAD and forms stable direct hydrogen bonds *via* the OH and indirectly to two residues Gln57 (2.3 Å) and Lys296 (2.5 Å) (figure 4, top). The scaffold binds to the residues Gln206, Tyr326, Thr201 and Ile199. The methyl group is well tolerated through hydrophobic

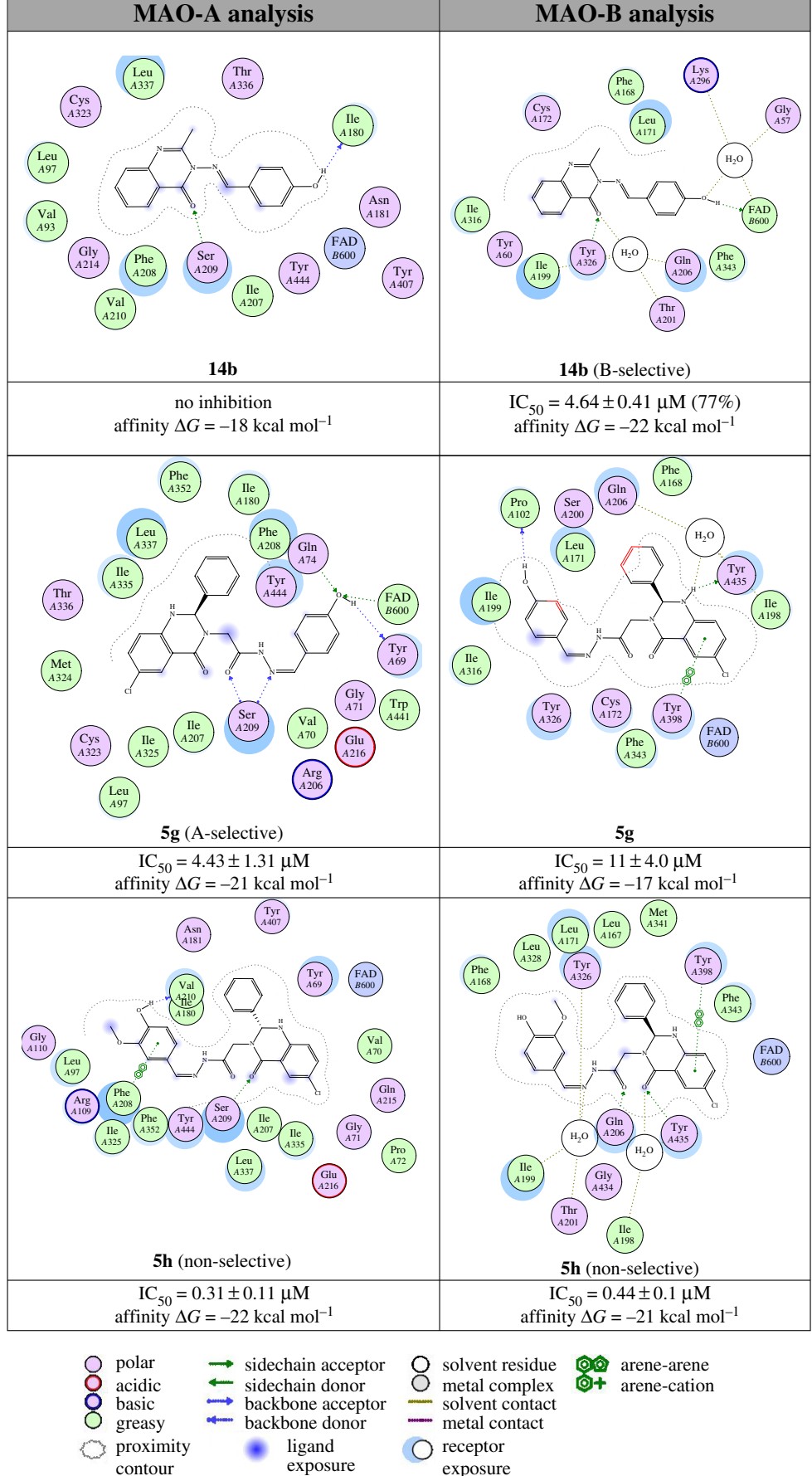

**Figure 4.** Interactions of selective and non-selective inhibitors with MAO-A and MAO-B.

**Table 2.** Detailed docking data of representative compounds with MAO-A and MAO-B.

| ID | fragment | MAO-A | MAO-B | interaction type |
|----|----------|-------|-------|------------------|
| **14b** | methyl quinazoline head | hydrophobic interaction with terminal pocket formed of Gly214, Phe208, Val93, Leu97, Lys323 and Thr336 | hydrophobic interaction with terminal pocket formed of Tyr80, Ile316, Cys172 and Ile199 | hydrophobic and hydrogen bonding |
| | | hydrogen bonding through the ($C=O$) to Ser209 with 2.2 Å | Bidirectional hydrogen bonding through the ($C=O$) to Tyr325 with 2.4 Å and $H_2O$-Thr201, with 4.6 Å | |
| | Imino linker | Vdw interaction with terminal pocket formed of Ser209 | Vdw interaction with terminal pocket formed of Leu171 | Vdw interaction |
| | aromatic ring tail | aromatic stacking tolerated in cage formed of Phe208 and FAD | hydrophobic interaction with terminal pocket formed of Gly206 and Phe343 | hydrophobic and hydrogen bonding |
| | | hydrogen bonding through the ($p$-OH) to Ile180 with 2.7 Å | bidirectional hydrogen bonding through the ($p$-OH) to FAD with 2.6 Å and $H_2O$-Gly57, with 4.5 Å | |
| **5g** | Phenyl dihydroquinazoline head | hydrophobic interaction with terminal pocket formed of Leu337, Ile207, Ile325 and Phe208 | aromatic stacking tolerated in cage formed of Tyr398 | hydrophobic and hydrogen bonding |
| | | | hydrogen bonding through (NH) fragment to Tyr435 with distance 2.3 Å | |
| | acetohydrazide linker | bidirectional hydrogen bonding through the (C=O) and (=N) to Ser209 with 1.95 and 2.20 Å | hydrophobic interaction with terminal pocket formed of Tyr326 and Cys172 | hydrophobic and hydrogen bonding |
| | aromatic ring tail | bidirectional hydrogen bonding through the ($p$-OH) to Gly74 and Tyr69 with 1.85, 2.9 Å | hydrogen bonding through the ($p$-OH) to Pro102 with 2.51 Å | hydrophobic and hydrogen bonding |
| | | hydrophobic interaction with terminal pocket formed of Gly71, Tyr69, Tyr444 and FAD | hydrophobic interaction with terminal residue Ile199, Leu171 and Ile316 | |
| **5h** | Phenyl dihydroquinazoline head | hydrophobic interaction with terminal pocket formed of Tyr69, Ile207, Ile335, Tyr407 and Val70 | hydrophobic interaction with terminal pocket formed of Phe343, Tyr435 and FAD | hydrophobic interaction |
| | | hydrogen bonding through the ($C=O$) to Ser209 with 2.7 Å | hydrogen bonding through the ($C=O$) to Tyr435 with 2.3 Å and $H_2O$-Ile198 with 4.9 Å | hydrophobic and hydrogen bonding |
| | | | aromatic stacking tolerated in cage formed of Tyr398 | |
| | acetohydrazide linker | aromatic stacking tolerated in cage formed of Phe208 | hydrogen bonding through the ($C=O$) to Gln206 with 3.1 Å | hydrophobic and hydrogen bonding |
| | aromatic ring tail | hydrogen bonding through the ($p$-OH) to Val210 with 2.4 Å | hydrophobic interaction with terminal pocket formed of Phe168, Leu328 and Tyr326 | hydrophobic and hydrogen bonding |
| | | Vdw interaction with terminal pocket formed of Leu97 | | |

interaction with Phe168 and Leu171. By contrast, the docking of **14b** (E-form) in the MAO-A formed unfavourable interactions in the pocket. The binding analysis of the MAO-A selective **5g** (Z-form) showed favourable interaction in the pocket with stable hydrogen bonding interactions through the terminal tail OH with Tyr69, FAD and Gln74 (figure 4, middle). It also binds through the acetohydrazide linker to the Ser209 by dual hydrogen bonds (2.72 and 2.48 Å). The phenylquinazoline head had hydrophobic interaction with the pocket formed of Phe352, Ile180, Leu337 and Phe208 residues. By contrast, MAO-B with **5g** (Z-form) showed serious clashes with both head and tail parts, and reversed placement compared to **14b**. The non-selective compound **5h** revealed the head phenylquinoxaline close to the FAD and good interaction behaviour towards the both targets (figure 4, bottom). All molecular placement data of active compounds within the binding pockets of MAO-A compared to MAO-B are summarized in table 2.

## 2.4. SAR analysis

MAO-A is inhibited only by the quinazoline compounds of type **5** (**5a**, **5d**, **5g**, **5h**) and with only micromolar potency, in contrast with our earlier 3-benzylquinoxaline-based compounds found to have nM potency [24]. For MAO-B, compounds **5a**, **5d**, **5g**, **5h** and **12d** gave full inhibition with micromolar affinity. The **14** series all gave 75% inhibition in the dose–response curves and $IC_{50}$ values about 6 µM. None of the series **14** compounds inhibited MAO-A, so these are selective MAO-B inhibitors. Interestingly, MAO-A and MAO-B were equally inhibited by the precursor **11**, and by **12b**, **5d** and **5h**, all with substituted aryl groups as the aromatic tail.

Considering first the phenylquinazoline head shown in the design strategy in chart 1 (type **5** compounds), the addition of Cl to the head group had no effect. However, switching the 2-methyl in **12** to the 2-phenyl in **5** improved inhibition of both MAO-A and MAO-B activity, from no inhibition to $IC_{50}$ values below 10 µM (table 1). Changing the linker from the acetylhydrazide found in **5** and **12** to the imine found in **14** introduced selectivity for MAO-B. Both **5a** and **14a** have $IC_{50}$ values below 10 µM for MAO-B but **5a** inhibits MAO-A with an $IC_{50}$ of 4.4 µM, whereas **14a** does not inhibit MAO-A. In the third region, the addition of 4-OH and 3-$OCH_3$ to the aromatic tail improves the $IC_{50}$ by more than 10-fold (**5a** with MAO-B is 10 µM, but **5d** is 0.75 µM; 6.2 and 0.27 µM for MAO-A). By contrast, with only the 4-OH, compounds **5b** and **5g** are poor inhibitors. This difference might be due to the changed interactions in the active site (figure 4 and table 2).

# 3. Conclusion

Prior work showed that the 3-benzylquinoxaline and quinazoline scaffolds gave a good basis for selective inhibition of MAO-A. Here, compounds were prepared to explore different synthetic methodologies and study the influence of the heterocyclic head group, the hydrazido linker and the aromatic tail in the inhibition. The N′-substituted benzylidene-2-(6-chloro-4-oxo-2-aryl-1,4-dihydroquinazolin-3(2H)-yl)aceto-hydrazide series provided the most potent inhibitors, namely **5d** ($IC_{50}$ = 0.25 µM, threefold selective for MAO-A) and **5h** (0.31 µM on MAO-A and 0.44 µM on MAO-B). Removing the aryl tail substituents resulted in decreased inhibition (**5a** $IC_{50}$ = 6 µM). The shorter less flexible linker in (E)-3-(substituted benzylideneamino)-2-methylquinazolin-4(3H)-one compounds **14a–14d** resulted in inhibition selective for MAO-B with $IC_{50}$ values in the micromolar range. Molecular docking revealed the importance of interactions near the FAD for good affinity.

# 4. Experimental section

## 4.1. General chemistry

Melting points were determined with a Mel-Temp apparatus and are uncorrected. Nuclear magnetic resonance spectra ($^1$H NMR and $^{13}$C NMR spectra) were recorded using a Bruker 400 MHz spectrometer with the chemical shift values reported in $\delta$ units (part per million) and the coupling constant (J) in hertz. Infrared data were obtained using a Perkin–Elmer 1600 series Fourier transform instrument as KBr pellets. Mass spectra were recorded on Brucker MALDI microflex and Agilent 6320 Ion Trap equipped with an electrospray ionization interface mass spectrometer. Elemental analyses were performed on Perkin–Elmer 2400 elemental analyser, and the values found were within ±0.3% of the theoretical values. The follow-up of the reactions and checking the purity of the compounds were

made by TLC on silica gel-protected aluminium sheets (Type 60 GF254, Merck) and the spots were detected by exposure to UV-lamp at $\lambda$ 254 nm for few seconds. Compounds of type **12** [31,32] and **14** were prepared according to the reported procedure [34–38] (electronic supplementary material).

### 4.1.1. General procedure for the preparation of methyl (2-aminobenzoyl)-glycinate (2a–2b)

In round bottom flask (500 ml), a mixture of glycine methyl ester hydrochloride (1.26 g, 10 mmol), triethylamine (1 ml) was dissolved in ethanol (300 ml) and heated in water both at 60–70°C for 7 min, isatoic anhydride (10 mmol) was added to the mixture (portion wise) over a period of 30 min with stirring at 80°C. The mixture was allowed to heat under reflux for extra 30 min. The solvent was evaporated and the residue was used in the next step without further purification.

#### 4.1.1.1. Methyl (2-aminobenzoyl)glycinate (2a) [25]
Yield: 69%; mp 75–77°C.

#### 4.1.1.2. Methyl (2-amino-5-chlorobenzoyl)glycinate (2b)
Yield: 71%; mp 155–157°C. IR (KBr): 3240, 3182 ($NH_2$ and NH), 1718 (C=O, ester), 1670 (C=O, amide) $cm^{-1}$.

### 4.1.2. General procedure for the preparation of methyl 2-(4-oxo-2-phenyl-1,2-dihydroquinazolin-3(4H)-yl)acetate (3a–3d)

Method A: To the solution of methyl (2-amino-5-substituted benzoyl)glycinate 2 (10 mmol) in ethanol (50 ml) was added the appropriate aldehydes and four drops of acetic acid at room temperature. The reaction mixture was heated under reflux for about 8 h and followed by TLC (EtOAc : MeOH 7 : 3). The solvent was removed under vacuum and the crude product was recrystallized from ethanol, filtered and washed with ethanol to afford the pure product.

Method B: Isatoic anhydride (6.13 mmol), triethylamine (0.619 g, 6.13 mmol), glycine methyl ester hydrochloride (0.769 g, 6.13 mmol), the appropriate aldehyde (6.13 mmol) and CuO nanoparticles (0.042 g, 0.613 mmol) were mixed in aqueous ethanol (EtOH : $H_2O$ 15 : 5) (20 ml). The reaction mixture was heated under reflux for 8 h and followed by TLC (EtOAc : MeOH 7 : 3). The CuO nanoparticle catalyst was separated by filtration and the filtrate was concentrated and left to cool to room temperature. The product that precipitated out was filtered off and recrystallized from ethanol.

#### 4.1.2.1. Methyl-2-(4-oxo-2-phenyl-1,2-dihydroquinazolin-3(4H)-yl)acetate (3a)
Method A: yield: 79%; Method B: yield: 83%; mp 198–200°C [25]

#### 4.1.2.2. Methyl-2-(6-chloro-4-oxo-2-phenyl-1,2-dihydroquinazolin-3(4H)-yl)acetate (3b)
Method A: yield: 77%; Method B: yield: 91%: mp 162–163°C; IR (KBr): 3375 (NH), 1750 (C=O, ester), 1630 (C=O, amide) $cm^{-1}$; [1]H NMR (400 MHz: DMSO-$d_6$): $\delta$ 3.60 (s, 3H, $OCH_3$), 3.63 (d, 1H, J 17.2 Hz, $\alpha$-CH), 4.41 (d, 1H, J 17.2 Hz, $\alpha$-CH), 5. 98 (s, 1H, $sp^3$ C-H), 6.74 (d, 1H, J 8 Hz, Ar-H), 7.31 (d, 1H, J 8 Hz, Ar-H), 7.40 (bs, 5H, Ar-H), 7.59 (bs, 1H, NH), 7.63 (s, 1H, Ar-H). [13]C NMR (100 MHz: DMSO- $d_6$): $\delta$ 46.32, 52.36, 71.95, 115.43, 116.89, 121.37, 127.07, 127.32, 129.23, 129.59, 133.99, 139.81, 146.39, 162.23, 169.46. Anal. Calcd for $C_{17}H_{15}ClN_2O_3$: C, 61.73; H, 4.57; N, 8.47. Found: C, 62.46; H, 4.61; N, 8.45.

#### 4.1.2.3. Methyl 2-(6-chloro-2-(4-methoxyphenyl)-4-oxo–1,2-dihydroquinazolin-3(4H)-yl)acetate (3c)
Method B: yield: 55%: mp 148–150°C. IR (KBr): 3300 (NH), 1710 (C=O, ester), 1630 (C=O, amide) $cm^{-1}$. [1]H NMR (400 MHz: DMSO-$d_6$): $\delta$ 3.58 (d, 1H, J 17.2 Hz, $\alpha$-CH), 3.60 (s, 3H, $OCH_3$), 3.75 (s, 3H, $OCH_3$), 4.34 (d, 1H, J 17.2 Hz, $\alpha$-CH), 5.92 (s, 1H, $sp^3$ C-H), 6.75 (d, 1H, J 8.4 Hz, Ar-H), 6.96 (d, 2H, J 8.2 Hz, Ar-H), 7.32 (d, 1H, J 8.4 Hz, Ar-H), 7.34 (d, 2H, J 8.2 Hz, Ar-H), 7.53 (bs, 1H, NH), 7.58 (s, 1H, Ar-H). [13]C NMR (100 MHz: DMSO$d_6$): $\delta$ 46.00, 52.35, 55.66, 71.72, 114.53, 115.37, 116.90, 121.54, 127.04, 129.08, 131.41, 134.06, 146.75, 160.29, 162.45, 169.47. Anal. Calcd for $C_{18}H_{17}ClN_2O_4$: C, 59.92; H, 4.75; N, 7.76. Found: C, 60.01; H, 4.61; N, 7.55.

### 4.1.2.4. Methyl 2-(6-chloro-2-(2-methoxyphenyl)-4-oxo–1,2-dihydroquinazolin-3(4H)-yl)acetate (3d)

Method B: yield: 55%: mp 152–154°C. IR (KBr): 3300 (NH), 1751 (C=O, ester), 1639 (C=O, amide) cm$^{-1}$. $^1$H NMR (400 MHz: DMSO-d$_6$): $\delta$ 3.60 (s, 3H, OCH$_3$), 3.62 (d, 1H, J 17.2 Hz, $\alpha$-CH), 3.79 (s, 3H, OCH$_3$), 4.41 (d, 1H, J 17.2 Hz, $\alpha$-CH), 6.22 (s, 1H, sp$^3$ C-H), 6.76 (d, 1H, J 7.6 Hz, Ar-H), 6.93 (t, 1H, J 7.6 Hz, Ar-H), 7.07 (d, 1H, J 7.6 Hz, Ar-H), 7.20 (d, 1H, J 7.6 Hz, Ar-H), 7.27 (d, 1H, J 7.6 Hz, Ar-H), 7.28 (bs, 1H, NH), 7.34 (t, 1H, J 7.6 Hz, Ar-H), 7,58 (s, 1H, Ar-H). $^{13}$C NMR (100 MHz: DMSO-d$_6$): $\delta$ 44.82, 51.04, 54.84, 65.44, 111.04, 114.05, 115.60, 119.79, 119.94, 125.85, 126.12, 126.24, 129.62, 132.60, 145.40, 156.09, 161.23, 168.43. Anal. Calcd for C$_{18}$H$_{17}$ClN$_2$O$_4$: C, 59.92; H, 4.75; N, 7.76. Found: C, 60.14; H, 4.73; N, 7.62.

### 4.1.3. General procedures for the preparation of 2-(6-substituted-4-oxo-2-phenyl-1,2 dihydroquinazolin-3(4H)-yl)acetohydrazide (4a–4d)

To a solution of compound 3 (3.1 mmol) in methanol (10 ml), hydrazine hydrate (80%) (0.4 ml) was added. The reaction mixture was heated under reflux for 3 h and monitored by TLC (EtOAc : MeOH 7 : 3). The solution was concentrated under vacuum and the product that precipitated out was filtrated off, recrystallized from ethanol and dried.

### 4.1.3.1. 2-(4-Oxo-2-phenyl-1,2-dihydroquinazolin-3(4H-yl)acetohydrazide (4a)

Yield 77%; mp 215–218°C [25].

### 4.1.3.2. 2-(6-Chloro-4-oxo-2-phenyl-1,2-dihydroquinazolin-3(4H)-yl)acetohydrazide (4b)

Yield 75%; mp 184–185°C; IR (KBr): 3320, 3300 (NH$_2$ and NH), 1675 (C=O, amide) cm$^{-1}$. $^1$H NMR (400 MHz: DMSO-d$_6$): $\delta$ 3.18 (d, J 16 Hz, 1H, $\alpha$-CH), 4.20 (s, 2H, NH$_2$), 4.5 (d, J 16 Hz, 1H, $\alpha$-CH), 5.96 (s, 1H, sp3 CH) 6.72 (d, J 8 Hz, 1H, Ar-H), 7.28 (d, J 8 Hz, 1H Ar-H), 7.39 (bs, 5H Ar-H), 7.52 (s, 1H, NH), 7.59 (s, 1H, Ar-H), 9.052 (s, 1H, NHCO). $^{13}$C NMR (100 MHz: DMSO-d$_6$): 45.62, 71.98, 115.72, 116.75, 121.22, 127.08, 127.27, 129.2, 129.50, 133.73, 140.02, 146.16, 162.24, 167.45. Anal. Calcd for C$_{16}$H$_{15}$ClN$_4$O$_2$: C, 58.10; H, 4.57; N, 16.94. Found: C, 57.95; H, 4.61; N, 16.81.

### 4.1.3.3. 2-(6-Chloro-2-(4-methoxyphenyl)-4-oxo-1,2-dihydroquiazolin-3(4H)-yl)acetohydrazide (4c)

Yield 86%; mp 159–160°C; IR (KBr): 3310, 3230 (NH$_2$ and NH), 1690 (C=O, amide) cm$^{-1}$. $^1$H NMR (400 MHz: DMSO-d$_6$): $\delta$ 3.56 (d, J 17.2, 1H, $\alpha$-CH), 3.60 (s, 3H, NH$_2$ and NH), 3.75 (s, 3H, OCH$_3$), 4.34 (d, J 17.2 HZ, 1H, $\alpha$-CH), 5.92 (s, 1H, sp$^3$ CH), 6.72 (d, J 8.4 Hz, 1H, Ar-H), 6.96 (d, J 8 Hz, 2H, Ar-H), 7.31 (d, J 8.4 Hz, 1H Ar-H), 7.34 (d, J 8 Hz, 2H, Ar-H), 7.49 (bs, 1H, NHCO), 7.58 (s, 1H, Ar-H). $^{13}$C NMR (DMSO-d$_6$, 100 MHz): $\delta$ 51.26, 54.60, 70.72, 113.47, 114.39, 115.83, 120.31, 126.02, 127.84, 130.36, 132.85, 145.53, 159.29, 161.34, 168.44. Anal. Calcd for C$_{17}$H$_{17}$ClN$_4$O$_3$: C, 56.59; H, 4.75; N, 15.53. Found: C, 56.35; H, 4.72; N, 15.65.

### 4.1.3.4. 2-(6-Chloro-2-(2-methoxyphenyl)-4-oxo-1,2-dihydroquinazolin-3(4H)-yl)acetohydrazide (4d)

Yield 87%; mp 149–151°C. IR (KBr): 3310, 3200 (NH$_2$ and NH), 1680 (C=O, amide) cm$^{-1}$. $^1$H NMR (400 MHz: DMSO-d$_6$): $\delta$ 3.60 (s, 3H, NH, NH$_2$), 3.62 (d, J 17.2 Hz, 1H, $\alpha$-CH), 3.79 (s, 3H, OCH$_3$), 4.35 (d, J 17.2 Hz, 1H,$\alpha$-CH), 6.22 (s, 1H, sp$^3$ CH), 6.76 (d, J 8 Hz, 1H, Ar-H), 6.93 (t, 2H, J 7.2 Hz, Ar-H), 7.07 (d, J 8 Hz, 1H, Ar-H), 7.20 (d, J 7.2 Hz, 1H, Ar-H), 7.26–7.28 (m, 2H, Ar-H, NHCO), 7.34 (t, J 8 Hz, 1H, Ar-H), 7.57 (s, 1H, Ar-H). $^{13}$C NMR (DMSO-d$_6$, 100 MHz): $\delta$ 52.30, 56.21, 66.75, 111.94, 115.13, 116.84, 120.86, 121.03, 126.91, 127.19, 127.31, 130.75, 133.80, 146.70, 157, 162.25, 169.41. Anal. Calcd for C$_{17}$H$_{17}$ClN$_4$O$_3$: C, 56.59; H, 4.75; N, 15.53. Found: C, 56.43; H, 4.75; N, 15.57.

### 4.1.4. General procedures for the preparation of N′-substituted benzylidene-2-(4-oxo-2-phenyl-1,4-dihydroquinazolin-3(2H)-yl)acetohydrazide (5a–5h)

To a solution of compound 4 (0.25 g, 6.85 mmol) in ethanol (20 ml), the appropriate aldehyde (6.85 mmol) was added and the mixture was stirred for 24 h at temperature between 25 and 60°C and monitored by TLC. When the reaction is over, the mixture was concentrated and left to cool to room temperature. The product that precipitated out was filtrated off, recrystallized from ethanol and dried.

### 4.1.4.1. N′-Benzylidene-2-(4-oxo-2-phenyl-1,2-dihydroquinazolin-3(4H)-yl)acetohydrazide (5a)

Yield: 88%; mp 223–225°C. IR (KBr): 3304 (NH), 1689, 1640 (C=O, amide) cm$^{-1}$. Isomer E (70%): $^1$H NMR (400 MHz: DMSO-d$_6$): $\delta$ 3.66 (d, 1H, J = 17.2 Hz, $\alpha$-CH), 5.05 (d, 1H, J = 17.2 Hz, $\alpha$-CH), 5.94 (s, 1H, sp$^3$ C-H), 6.70–6.72 (m, 2H, Ar-H), 7.25–7.69 (m, 13H, Ar-H and NH), 7.93 (s, 1H, CH = N), 11.49 (s, 1H, NHCO). $^{13}$C NMR (100 MHz: DMSO-d$_6$): $\delta$ 45.47, 72.22, 114.73, 114.79, 117.67, 127.19, 127.58, 128.11, 129.20, 129.29, 130.34, 133.94, 134.35, 140.29, 143.95, 147.04, 147.74, 163.56, 169.54. Isomer Z (30%): $^1$H NMR (400 MHz: DMSO-d$_6$): $\delta$ 3.12 (d, 1H, J = 16.8 Hz, $\alpha$-CH), 4.37 (d, 1H, J = 16.8 Hz, $\alpha$-CH), 5.92 (s, 1H, sp$^3$ C-H), 6.70–6.72 (m, 2H, Ar-H), 7.25–7.69 (m, 13H, Ar-H and NH), 8.15 (s, 1H, CH = N), 11.44 (s, 1H, NHCO). $^{13}$C NMR (100 MHz: DMSO-d$_6$): $\delta$ 46.40, 72.33, 114.57, 114.68, 117.59, 127.40, 127.45, 127.50, 128.09, 129.17, 129.48, 130.50, 133.89, 134.06, 134.64, 147.53, 147,59, 163.59, 164.84. MS-MALDI: m/e 385.06 [M+1]. Anal. Calcd for C$_{23}$H$_{20}$N$_4$O$_2$: C, 71.86; H, 5.24; N, 14.57. Found: C, 71.02; H, 4.95; N, 14.41.

### 4.1.4.2. N′-(4-Hydroxybenzylidene)-2-(4-oxo-2-phenyl-1,2-dihydroquinazolin-3(4H)-yl)acetohydrazide (5b)

Yield: 84%; mp 230–232°C. IR (KBr): 3275 (br) (OH and NH), 1685, 1650 (C=O, amide) cm$^{-1}$. Isomer E (70%): $^1$H NMR (400 MHz: DMSO-d$_6$): $\delta$ 3.60 (d, 1H, J 17.2 Hz, $\alpha$-CH), 5.03 (d, IH, J 17.2 Hz, $\alpha$-CH) 5.99 (s, 1H, sp$^3$ C-H), 6.71, 6.78, 7.27, 7.37, 7.67 (m, d, m, d, d, 14H, Ar-H and NH) 7.82 (s, 1H, CHN), 9.89 (s, 1H, OH), 11.28 (s, IH, NHCO). $^{13}$C NMR (100 MHz: DMSO-d$_6$): $\delta$ 45.43, 72.23, 114.71, 114.80, 116.14, 117.64, 125.38, 127.57, 128.10, 128.91, 129.20, 129.47, 133.92, 140.32, 144.24, 147.72, 159.65, 163.55, 169.18. Isomer Z (30%): $^1$H NMR (400 MHz: DMSO-d$_6$): $\delta$ 3.33 (d, IH, J=16.4 Hz, $\alpha$-CH), 4.52 (d, IH, J=17.2 Hz, $\alpha$-CH), 5.99 (s, 1H, sp$^3$ C-H), 6.71, 6.82, 7.27, 7.44, 7.51, (m, d, m, d, d, 14H, Ar-H and NH), 8.03 (s, 1H, CHN), 9.93 (s, 1H, OH), 11.21 (s, 1H, NHCO). $^{13}$C NMR (100 MHz: DMSO-d$_6$): $\delta$ 46.30, 72.29, 114.60, 114.74, 117.64, 125.59, 127.43, 128.10, 129.20, 129.26, 129.45, 134.04, 140.32, 144.24, 147.35, 147.57, 159.83, 164.44, 169.18. MS-ESI: m/e 399.6 [M+1–2]. Anal. Calcd for C$_{23}$H$_{20}$N$_4$O$_3$: C, 68.99; H, 5.03; N, 13.99. Found: C, 68.75; H, 4.79; N, 13.61.

### 4.1.4.3. N′-(4-Methoxybenzylidene)-2-(4-oxo-2-phenyl-1,2-dihydroquinazolin-3(4H)-yl)acetohydrazide (5c)

Yield: 91%; mp 210–212°C. IR (KBr): 3300, 3290 (NH), 1690 (C=O, amide) cm$^{-1}$. Isomer E (70%): $^1$H NMR (400 MHz: DMSO-d$_6$): $\delta$ 3.60 (d, 1H, J 17.2 Hz, $\alpha$-CH), 5.03 (d, IH, J 17.2 Hz, $\alpha$-CH) 5.99 (s, 1H, sp$^3$ C-H), 6.71, 6.78, 7.27, 7.37, 7.67, (m, d, m, d, d,14H, Ar-H and NH) 7.82 (s, 1H, CHN), 9.89 (s, 1H, OH), 11.28 (s, IH, NHCO). $^{13}$C NMR (100 MHz: DMSO-d$_6$): $\delta$ 45.43, 72.23, 114.71, 114.80, 116.14, 117.64, 125.38, 127.57, 128.10, 128.91, 129.20, 129.47, 133.92, 140.32, 144.24, 147.72, 159.65, 163.55, 169.18. Isomer Z (30%): $^1$H NMR (400 MHz: DMSO-d$_6$): $\delta$ 3.33 (d, IH, J 16.4 Hz, $\alpha$-CH), 4.52 (d, IH, J 16.4 Hz, $\alpha$-CH), 5.99 (s, 1H, sp$^3$ C-H), 6.71, 6.82, 7.27, 7.44, 7.51, (m, d, m, d, d, 14H, Ar-H and NH), 8.03 (s, 1H, CHN), 9.93 (s, 1H, OH), 11.21 (s, IH, NHCO). $^{13}$C NMR (100 MHz: DMSO-d$_6$): $\delta$ 46.30, 72.29, 114.60, 114.74, 117.64, 125.59, 127.43, 128.10, 129.20, 129.26, 129.45, 134.04, 140.32, 144.24, 147.35, 147.57, 159.83, 164.44, 169.18. Anal. Calcd for C$_{24}$H$_{22}$N$_4$O$_3$: C, 69.55; H, 5.35; N, 13.52. Found: C, 70.05; H, 5.42; N, 13.65.

### 4.1.4.4. N′-(4-Hydroxy-3-methoxybenzylidene)-2-(4-oxo-2-phenyl-1,2-dihydroquinazolin-3(4H)yl)acetohydrazide (5d)

Yield: 89%; mp 252–255°C. IR (KBr): 3300, 3295, 3200 (OH and NH), 1700 (C=O, amide) cm$^{-1}$. Isomer E (64%): $^1$H NMR (400 MHz: DMSO-d$_6$): $\delta$ 3.64 (d,1H, J 17.6 Hz, $\alpha$-CH), 3.76 (s, 3H, OCH$_3$), 4.03 (d, 1H, J 17.6 Hz, $\alpha$-CH), 6.00 (s, 1H, sp$^3$ C-H), 6.70, 6.69, 7.00, 7.25–7.49, 7.68 (m, d, d, s, m, d, 13H, ArH and NH), 7.82 (s, 1H, CHN), 9.50 (s, 1H, OH), 11.32 (s, 1H, NHCO). $^{13}$C NMR (100 MHz: DMSO-d$_6$): $\delta$ 45.42, 55.98, 72.25, 110.06, 114.73, 114.82, 116.03, 117.69, 121.40, 125.81, 127.62, 128.12, 129.17, 129.21, 129.48, 133.93, 140.25, 144.39, 147.77, 148.33, 149.14, 163.58, 169.25. Isomer Z (36%): $^1$H NMR (400 MHz: DMSO-d$_6$): $\delta$ 3.34 (d, 1H, J 16.4 Hz, $\alpha$-CH), 3.81 (s, 3H, OCH$_3$), 4.53 (d, 1H, J 16.4 Hz, $\alpha$-CH), 6.00 (s, 1H, sp$^3$ C-H), 6.70, 6.83, 7.06, 7.25–7.49, 7.68 (m, d, d, m, d, 13H, Ar-H + NH), 8.03 (s, 1H, CHN), 9.55 (s, 1H, OH), 11.25 (s, 1H, NHCO). $^{13}$C NMR (100 MHz: DMSO-d$_6$): $\delta$ 46.32, 56.52, 72.35, 109.51, 114.62, 114.76, 115.88, 117.69, 122.41, 126.01, 127.62, 127.48, 128.12, 129.17, 129.21, 129.48, 134.06, 140.28, 147.60, 148.45, 149.38, 163.62, 164.49. MS-ESI: m/e 429.0 [M+1–2]. Anal. Calcd for C$_{24}$H$_{22}$N$_4$O$_4$: C, 66.97; H, 5.15; N, 13.02. Found: C, 67.05; H, 4.99; N, 13.14.

### 4.1.4.5. N′-Benzylidene-2-(6-chloro-4-oxo-2-phenyl-1,4-dihydroquinazolin-3(2H)-yl)acetohydrazide (5e)

Yield: 92%; mp 266–270°C. IR (KBr): 3390, 3224 (NH), 1699, 1637 (C=O, amide) cm$^{-1}$. Isomer E (73%): $^1$H NMR (400 MHz: DMSO-d$_6$): $\delta$ 3.71 (d, 1H, J 17.2 Hz, $\alpha$-CH), 5.04 (d, IH, J 17.2 Hz, $\alpha$-CH), 6.02 (s, 1H, sp$^3$

C-H), 6.75, 7.31, 7.27, 7.43, 7.57, 7,69 (d, d, m, m, d, 15H, Ar-H and NH), 7.99 (s, 1H, CHN), 11.51 (s, IH, NHCO). $^{13}$C NMR (100 MHz: DMSO-d$_6$): $\delta$ 45.63, 72.11, 115.82, 116.79, 121.28, 127.20, 127.51, 129.26, 130.34, 133.73, 134.33, 139.96, 144.07, 146.45, 162.41, 169.27. Isomer Z (27%): $^1$H NMR (400 MHz: DMSO-d$_6$): $\delta$ 3.63 (d, IH, I 16.4 Hz, $\alpha$-CH), 4.52 (d, IH, J 16.42 Hz, $\alpha$-CH), 6.02 (s, 1H, sp$^3$ C-H), 6.75, 7.31, 7.27, 7.43, 7.57, 7,69 (d, d, m, m, d, 15H, Ar-H and NH), 8.15 (s, 1H, CH = N), 11.44 (s, 1H, NHCO). $^{13}$C NMR (100 MHz: DMSO-d$_6$): $\delta$ 46.47, 72.23, 115.59, 116.79, 127.11, 127.40, 129.26, 129.59, 130.51, 133.85, 134.62, 146.32, 147.19, 162.44, 169.24. MS-ESI: m/e 417 [M+1–2]. Anal. Calcd for C$_{23}$H$_{19}$ClN$_4$O$_2$: C, 65.95; H, 4.57; N, 13.38. Found: C, 66.02; H, 4.70; N, 13.52.

#### 4.1.4.6. 2-(6-Chloro-4-oxo-2-phenyl-1,4-dihydroquinazolin-3(2H)-yl)-N'-(4-methoxybenzylidene)-acetohydrazide (5f)

Yield: 90%; mp 138–140°C. IR (KBr): 3290, 3275 (NH), 1695, 1650 (C=O, amide) cm$^{-1}$. Isomer E (67%): $^1$H NMR (400 MHz: DMSO-d$_6$): $\delta$ 3.68 (d, 1H, J 17.2 Hz, $\alpha$-CH), 3.77 (s, 3H, CH$_3$O), 5.03 (d, 1H, J 17.2 Hz, $\alpha$-CH), 6.03 (s, 1H, sp$^3$ C-H), 6.76 (d, 1H, J 8.8 Hz, ArH), 6.95 (d, 2H, J 8.4 Hz, ArH), 7.31 (dd, 1H, J 8.4 Hz, 2.4 Hz, ArH), 7.38–7.64 (m, 9H, ArH and NH), 8.09 (s, 1H, CH = N), 11.32 (s, 1H, NHCO). $^{13}$C NMR (100 MHz: DMSO-d$_6$): $\delta$ 45.63, 55.10, 72.12, 114.74, 115.83, 116.78, 121.28, 126.94, 127.13, 127.52, 128.79, 129.26, 133.73, 139.96, 143.95, 146.45, 161.11, 162.42, 164.30, 169.02. Isomer Z (33%): $^1$H NMR (400 MHz: DMSO-d$_6$): $\delta$ 3.41 (d, 1H, J 16.4 Hz, $\alpha$-CH), 3.80 (s, 3H, OCH$_3$), 4.52 (d, 1H, J 16.4 Hz, $\alpha$-CH), 6.03 (s, 1H, sp$^3$ C-H), 6.95 (d, 1H, J 8.4 Hz, ArH), 7.00 (d, 2H, J 8.4 Hz, ArH), 7.31 (dd, 1H, J 8.4 Hz, 2.4 Hz, ArH), 7.38–7.64 (m, 9H, ArH and NH), 7.88 (s, 1H, CH = N), 11.39 (s, 1H, NHCO). $^{13}$C NMR (100 MHz: DMSO-d$_6$): $\delta$ 46.41, 56.53, 72.22, 114.74, 115.61, 116.81, 121.28, 126.94, 127.10, 127.40, 129.12, 129.60, 133.84, 139.96, 146.32, 147.08, 161.28, 162.45, 164.33, 169.02. Anal. Calcd for C$_{24}$H$_{21}$ClN$_4$O$_3$: C, 64.21; H, 4.72; N, 12.48. Found: C, 63.98; H, 4.84; N, 12.67.

#### 4.1.4.7. 2-(6-Chloro-4-oxo-2-phenyl-1,4-dihydroquinazolin-3(2H)-yl)-N'-(4-hydroxybenzylidene)acetohydrazide (5g)

Yield: 84%; mp 236–240°C. Isomer E (71%): $^1$H NMR (400 MHz: DMSO-d$_6$): $\delta$ 3.64 (d, 1H, J 17.2 Hz, $\alpha$-CH), 5.01 (d, 1H, J 17.2 Hz, $\alpha$-CH), 6.01 (s, 1H, sp$^3$ C-H), 6.74 (d, 1H, J 8.8 Hz, ArH), 6.78 (d, 2H, J 8.0 Hz, ArH), 7.31 (dd, 1H, J 8.8 Hz, 2.4 Hz, ArH), 7.37–7.57 (m, 9H, ArH and NH), 7.60 (d, 1H, J 2.4 Hz, ArH), 9.90 (br s, 1H, OH), 11.31 (s, 1H, NHCO). $^{13}$C NMR (100 MHz: DMSO-d$_6$): $\delta$ 45.61, 72.07, 115.81, 116.14, 116.77, 121.24, 125.35, 127.10, 127.49, 128.94, 129.27, 133.73, 140.00, 144.35, 146.43, 159.68, 162.37, 164.15, 168.89. Isomer Z (29%): $^1$H NMR (400 MHz: DMSO-d$_6$): $\delta$ 3.35 (d, 1H, J = 16.0 Hz, $\alpha$-CH), 4.50 (d, 1H, J 16.0 Hz, $\alpha$-CH), 6.01 (s, 1H, sp$^3$ C-H), 6.78 (d, 1H, J 8.0 Hz, ArH), 6.82 (d, 2H, J 8.4 Hz, ArH), 7.31 (dd, 1H, J 8.8 Hz, 2.4 Hz, ArH), 7.37–7.57 (m, 9H, ArH and NH), 7.60 (d, 1H, J 2.4 Hz, ArH), 9.90 (br s, 1H, OH), 11.23 (s, 1H, NHCO). $^{13}$C NMR (100 MHz: DMSO-d$_6$): $\delta$ 46.37, 72.15, 115.61, 116.14, 116.80, 121.24, 125.55, 127.25, 127.37, 129.27, 129.59, 133.84, 139.97, 146.29, 147.45, 159.85, 162.40, 164.15, 168.89. Anal. Calcd for C$_{23}$H$_{19}$ClN$_4$O$_3$: C, 63.52; H, 4.40; N, 12.88. Found: C, 63.76; H, 4.33; N, 13.15.

#### 4.1.4.8. 2-(6-Chloro-4-oxo-2-phenyl-1,4-dihydroquinazolin-3(2H)-yl)-N'-(4-hydroxy-3-methoxybenzylidene)-acetohydrazide (5h)

Yield: 93%; mp 240–242°C. Isomer E (67%): $^1$H NMR (400 MHz: DMSO-d$_6$): $\delta$ 3.67 (d, 1H, J 17.6 Hz, $\alpha$-CH), 3.76 (s, 3H, CH$_3$O), 5.01 (d, 1H, J 17.6 Hz, $\alpha$-CH), 6.01 (s, 1H, sp$^3$ C-H), 6.74 (d, 1H, J 8.8 Hz, ArH), 6.78 (d, 1H, J 8.0 Hz, ArH), 6.99 (dd, 1H, J 8.4 Hz, 1.2 Hz, ArH), 7.11 (d, 1H, J 1.2 Hz, ArH), 7.31 (dd, 1H, J 8.8 Hz, 2.4 Hz, ArH), 7.36–7.60 (m, 8H, ArH and NH), 9.52 (br s, 1H, OH), 11.33 (s, 1H, NHCO). $^{13}$C NMR (100 MHz: DMSO-d$_6$): $\delta$ 45.58, 55.98, 72.10, 110.09, 115.82, 116.02, 116.78, 121.26, 122.41, 125.77, 127.10, 127.54, 129.24, 129.60, 133.73, 139.92, 144.50, 146.47, 147.68, 148.33, 149.17, 162.40, 164.19, 168.95. Isomer Z (23%): $^1$H NMR (400 MHz: DMSO-d$_6$): $\delta$ 3.35 (d, 1H, J 16.4 Hz, $\alpha$-CH), 3.81 (s, 3H, CH$_3$O), 4.50 (d, 1H, J 16.4 Hz, $\alpha$-CH), 6.01 (s, 1H, sp$^3$ C-H), 6.74 (d, 1H, J 8.8 Hz, ArH), 6.82 (d, 1H, J 8.0 Hz, ArH), 6.99 (dd, 1H, J 8.4 Hz, 1.2 Hz, ArH), 7.26 (d, 1H, J 1.2 Hz, ArH), 7.31 (dd, 1H, J 8.8 Hz, 2.4 Hz, ArH), 7.36–7.60 (m, 8H, ArH and NH), 9.52 (br s, 1H, OH), 11.25 (s, 1H, NHCO). $^{13}$C NMR (100 MHz: DMSO-d$_6$): $\delta$ 46.37, 55.96, 72.20, 109.55, 115.62, 115.87, 116.81, 121.42, 122.41, 125.97, 127.10, 127.40, 129.29, 129.60, 133.84, 139.92, 144.50, 146.32, 147.68, 148.44, 149.40, 162.44, 164.19, 168.95. Anal. Calcd for C$_{24}$H$_{21}$ClN$_4$O$_4$: C, 62.00; H, 4.55; N, 12.05. Found: C, 61.81; H, 4.65; N, 12.13.

### 4.1.5. 2-(((E)-benzylidene)amino)-5-chloro-N-(2-(2-((E)-4-hydroxy-3-methoxybenzylidene)hydrazinyl)-2-oxoethyl)benzamide (6)

Yield: 36%; mp 257–259°C. [1]H NMR (400 MHz: DMSO-$d_6$): $\delta$ 3.64 (s, 2H, CH$_2$), 3.86 (s, 3H, CH$_3$O), 6.99 (d, 1H, J 7.2 Hz, ArH), 7.18 (d, 1H, J 8.8 Hz, ArH), 7.30 (s, 1H, ArH), 7.36 (d, 1H, J 7.2 Hz, ArH), 7.42 (s, 1H, ArH), 7.55–7.60 (m, 6H, ArH), 7.63 (s, 1H, ArH), 7.71 (s, 1H, ArH), 8.74 (s, 1H, NHCH$_2$), 8.80 (s, 1H, NHCO), 10.53 (s, 1H, OH). [13]C NMR (100 MHz: DMSO-$d_6$): $\delta$ 44.75, 56.05, 110.85, 116.05, 116.08, 123.10, 123.52, 124.63, 124.90, 127.53, 128.34, 130.40, 132.52, 136.62, 148.53, 148.58, 151.23, 151.83, 161.26, 166.88, 167.29, 171.30. Anal. Calcd for C$_{24}$H$_{21}$ClN$_4$O$_4$: C, 62.00; H, 4.55; N, 12.05. Found: C, 62.21; H, 4.71; N, 12.30.

### 4.1.6. Methyl 2-(4-oxo-2-phenylquinazolin-3(4H)-yl)acetate (8)

To a solution of compound 3a (0.75 g, 2.54 mmol) in ethyl acetate (50 ml), 2,3-dichloro-5,6-dicyano-p-benzoquinone DDQ (0.691 g, 3.048 mmol) was added. The reaction mixture was stirred for 3 h, concentrated, and left to cool to room temperature. The product was purified by column chromatography using silica gel as stationary phase and ethyl acetate–methanol (8 : 2) as mobile phase. Yield: 89%; mp 115–116°C [Lit. [29] 150°C]. IR (KBr): 1750 (C=O, ester), 1688 (C=O, amide) cm$^{-1}$. [1]H NMR (400 MHz: DMSO-$d_6$): $\delta$ 3.64 (s, 3H, OCH$_3$), 4.66 (s, 2H, CH$_2$), 7.57 (bs, 5H, Ar-H), 7.62 (t, 1H, J 8 Hz, Ar-H), 7.74 (d, 1H, J 8 Hz, Ar-H), 7.91 (t, 1H, J 8 Hz, Ar-H), 8.2 (d, 1H, J 8 Hz, Ar-H). [13]C NMR (100 MHz: DMSO-$d_6$): $\delta$ 47.86, 61.82, 120.32, 126.72, 127.90, 128.40, 129.17, 130.62, 134.95, 135.54, 147.35, 155.94, 161.62, 168.36, 168.87.

### 4.1.7. 2-(4-oxo-2-phenylquinazolin-3(4H)-yl)acetohydrazide (7)

To a solution of compound 8 (0.294 g, 1 mmol) in methanol (10 ml), hydrazine hydrate (80%) (0.4 ml) was added. The reaction mixture was heated under reflux for 3 h and monitored by TLC (EtOAc : MeOH 7 : 3). The solution was concentrated and the product that precipitated out was filtrated off, recrystallized from ethanol and dried. Yield: 87%; mp 215–218°C. IR (KBr): 3350, 3300 (NH$_2$ and NH), 1690 (C=O, amide) cm$^{-1}$.

### 4.1.8. Methyl 2-(2-methyl-4-oxoquinazolin-3(4H)-yl)acetate (10)

A mixture of compound 9 (1.03 g, 3 mmol), and triethylamine (0.636 g, 3 mmol), glycine methyl ester hydrochloride (0.803 g, 3 mmol). The reaction mixture was heated in an oil-bath at temperature of 120°C for 1 h, and the reaction was followed by TLC (EtOAc : MeOH 7 : 3). The crude residue was recrystallized from methanol and dried. Yield: 46%; mp 115–117°C [lit. 115°C [35]].

### 4.1.9. 2-(2-Methyl-4-oxoquinazolin-3(4H)-yl)acetohydrazide (11)

To a solution of compound 10 (0.5 g, 2.15 mmol) in dimethylformamide (10 ml), hydrazine hydrate (80%, 0.25 ml) and two drops of anhydrous acetic acid were added. The reaction mixture was heated under reflux for 5 h and followed by TLC (EtOAc : MeOH 7 : 3). The reaction mixture was left to cool to room temperature and the product that separated out was filtered off and recrystallized from ethanol, yield= (79%), mp 256°C decomp. [lit. [35] 260°C]. IR (KBr): 3267, 3142 (NH$_2$ and NH), 1670 (C=O, amide) cm$^{-1}$. [1]H NMR (400 MHz: DMSO-$d_6$): $\delta$ 2.50 (s, 3H, CH$_3$), 3.41 (bs, 2H, NH$_2$), 4.38 (s, 2H, CH$_2$), 7.49 (t, 1H, J 7.6 Hz, Ar-H), 7.61 (d, J 7.6 Hz, 1H, Ar-H), 7.81 (t, J 7.6 Hz, 1H, Ar-H), 8.08 (d, J 7.6 Hz, 1H, Ar-H), 8.83, 9.44 (s, bs, 1H, NH). [13]C NMR (100 MHz: DMSO-$d_6$): $\delta$ 22.26, 44.20, 119.13, 125.56, 125.65, 125.82, 133.83, 146.51, 154.77, 160.50, 165.70.

### 4.1.10. General procedures for the preparation of N′-substituted benzylidene-2-(2-methyl-4-oxoquinazolin-3(4H)-yl)acetohydrazide (12a–12d)

To a solution of compound 11 (1.624 g, 7.00 mmol) in dimethylformamide (10 ml), the appropriate aldehyde (7.00 mmol) and two drops of anhydrous acetic acid were added. The reaction mixture was heated under reflux for 5 h and the mixture was left to cool to room temperature. The product that precipitated out was filtrated off, recrystallized from ethanol and dried.

### 4.1.10.1. N′-Benzylidene-2-(2-methyl-4-oxoquinazolin-3(4H)-yl)acetohydrazide (12a)

Yield: 64%; mp 242–245°C [lit. [32] 245°C]. IR (KBr): 3223 (NH), 1680 (C=O, amide) cm$^{-1}$. Isomer E (80%): $^1$H NMR (400 MHz: DMSO-d$_6$): $\delta$ 2.56 (s, 3H, CH$_3$), 5.33 (s, 2H, CH$_2$), 7.46–8.12 (m, 10H, Ar-H), 11.85 (s, 1H, NH). $^{13}$C NMR (100 MHz: DMSOd$_6$): $\delta$ 23.31, 45.70, 120.08, 126.67, 126.84, 127.06, 127.45, 129.31, 130.59, 134.35, 135.01, 144.93, 147.66, 155.91, 161.69, 169.73. Isomer Z (20%): $^1$H NMR (400 MHz: DMSO-d$_6$): $\delta$ 2.51 (s, 3H, CH$_3$), 4.90 (s, 2H, CH$_2$), 7.46–8.25 (m, 10H, Ar-H), 11.90 (s, 1H, NH). $^{13}$C NMR (100 MHz: DMSOd$_6$): $\delta$ 23.46, 46.10, 120.10, 126.63, 126.87, 127.06, 127.62, 129.31, 130.66, 134.49, 135.04, 147.61, 147.75, 155.86, 164.00. MS-ESI: m/e 321.1 [M+1] (calcd for C$_{18}$H$_{16}$N$_4$O$_2$).

### 4.1.10.2. N′-(4-Hydroxybenzylidene)-2-(2-methyl-4-oxoquinazolin-3(4H)-yl)acetohydrazide (12b)

Yield: 86%; mp 192–195°C. IR (KBr): 3210, 3100 (br.) (OH and NH), 1680 (C=O, amide) cm$^{-1}$. Isomer E (77%): $^1$H NMR (400 MHz: DMSO-d$_6$): $\delta$ 2.56 (s, 3H, CH$_3$), 5.30 (s, 2H, CH$_2$), 6.83–6.87, 7.49–7.65, 7.80–7.84, 7.99–8.10–8.14 (m, m, t, s, m, 9H, Ar-H), 9.97 (s, 1H, OH), 11.65 (s, 1H, NH). $^{13}$C NMR (100 MHz: DMSO-d$_6$): $\delta$ 23.30, 45.66, 116.18, 120.09, 125.38, 126.67, 126.80, 127.04, 129.20, 134.97, 145.21, 147.66, 155.93, 159.89, 161.69, 168.32. Isomer Z (23%): $^1$H NMR (400 MHz: DMSO-d$_6$): $\delta$ 2.51 (s, 3H, CH$_3$), 4.88 (s, 2H, CH$_2$), 6.83–6.87, 7.49–7.65, 7.80–7.84, 7.99–8.10–8.14 (m, m, t, s, m, 9H, Ar-H), 9.97 (s, 1H, OH), 11.70 (s, 1H, NH). $^{13}$C NMR (100 MHz: DMSO-d$_6$): $\delta$ 23.45, 46.08, 116.18, 120.12, 125.45, 126.64, 126.84, 129.40, 135.00, 147.61, 148.04, 155.87, 159.99, 163.57. MS-ESI: m/e 337.0 [M+1] (calcd for C$_{18}$H$_{16}$N$_4$O$_3$). Anal. Calcd for C$_{18}$H$_{16}$N$_4$O$_3$: C, 64.28; H, 4.79; N, 16.66. Found: C, 64.50; H, 5.02; N, 16.84.

### 4.1.10.3. N′-(4-Methoxybenzylidene)-2-(2-methyl-4-oxoquinazolin-3(4H)-yl)acetohydrazide (12c)

Yield: 77%; M.P.: >229°C decomp. [lit. [32] 230°C] IR (KBr): 3400 (NH), 1690 (C=O, amide) cm$^{-1}$. Isomer E (78%): $^1$H NMR (400 MHz: DMSO-d$_6$): $\delta$ 2.52 (s, 3H, CH$_3$), 3.81 (s, 3H, OCH$_3$), 5.31 (s, 2H, CH$_2$), 7.02–7.07, 7.49–7.53, 7.62–7.71, 7.81–7.83, 8.03–8.19 (t, t, d, d, d, s, d, s, 9H, Ar-H), 11.72 (s, 1H, NH). $^{13}$C NMR (100 MHz: DMSOd$_6$): $\delta$ 23.30, 45.67, 55.77, 114.80, 120.09, 126.66, 126.82, 127.04, 129.06, 130.40, 134.99, 144.80, 147.66, 155.93, 161.06, 161.31, 161.69, 163.63. Isomer Z (22%): $^1$H NMR (400 MHz: DMSO-d$_6$): $\delta$ 2.56 (s, 3H, CH$_3$), 3.83 (s, 3H, OCH$_3$), 4.88 (s, 2H, CH$_2$), 7.02–7.07, 7.49–7.53, 7.62–7.71, 7.81–7.83, 8.03–8.19 (t, t, d, d, d, s, d, s, 9H, Ar-H), 11.77 (s, 1H, NH). $^{13}$C NMR (100 MHz: DMSOd$_6$): $\delta$ 23.40, 46.03, 55.84, 114.86, 120.12, 126.63, 126.85, 126.94, 129.23, 130.40, 135.02, 144.80, 147.61, 155.88, 161.06, 161.39, 162.14, 168.44. MS-ESI: m/e 350.7 [M+1].

### 4.1.10.4. N′-(4-Hydroxy-3-methoxybenzylidene)-2-(2-methyl-4-oxoquinazolin-3(4H)-yl)acetohydrazide (12d)

Yield: 72%; mp 258–260°C. IR (KBr): 3294 (br), 3180 (br) (OH and NH), 1699, 1686 (C=O, amide) cm$^{-1}$. Isomer E (81%): $^1$H NMR (400 MHz: DMSO-d$_6$): $\delta$ 2.61 (s, 3H, CH$_3$), 3.84 (s, 3H, OCH$_3$), 5.31 (s, 2H, CH$_2$), 6.85–8.23 (m, 8H, Ar-H), 9.47 (s, 1H, OH), 11.68 (s, 1H, NH). $^{13}$C NMR (100 MHz: DMSOd$_6$): $\delta$ 23.32, 45.29, 110.0 7, 116.01, 120.08, 120.21, 121.98, 125.78, 126.63, 126.75, 126.86, 127.06, 134.89, 145.36, 147.58, 148.45, 149.42, 155.91, 166.79, 168.23. Isomer Z (19%): $^1$H NMR (400 MHz: DMSO-d$_6$): $\delta$ 2.59 (s, 3H, CH$_3$), 3.81(s, 3H, OCH$_3$), 4.88 (s, 2H, CH$_2$), 6.85–8.23 (m, 8H, Ar-H), 9.58 (s, 1H, OH), 11.76 (s, 1H, NH). $^{13}$C NMR (100 MHz: DMSOd$_6$): $\delta$ 23.18, 45.73, 109.65, 115.91, 120.17, 122.49, 125.86, 126.65, 126.70, 126.83, 126.99, 135.00, 147.55, 148.45, 149.54, 166.15, 167.79. MS-ESI: m/e 336.4 [M+1]. Anal. Calcd for C$_{19}$H$_{18}$N$_4$O$_4$: C, 62.29; H, 4.95; N, 15.29. Found: C, 62.17; H, 5.01; N, 15.43.

## 4.2. MAO activity screening

The activity of MAO was determined using the coupled assay method of [41,42] in which the product, H$_2$O$_2$, acts with horseradish peroxidase to convert 10-acetyl-3,7-dihydroxyphenoxazine to the fluorescent resorufin. All compounds were screened for inhibition of the activity of membrane-bound human MAO-A and B at 10 μM in the presence of $2 \times K_m$ tyramine (an assay concentration of 0.8 mM with MAO-A and 0.32 mM with MAO-B). For compounds that gave more than 30% inhibition, the IC$_{50}$ was determined using seven concentrations of each compound (0.01–300 μM) with substrate at $2 \times K_M$. The IC$_{50}$ values are the mean and standard deviation for two separate experiments (16 and 22 points, respectively). The data were analysed using the equation for a three-parameter curve in Graphpad PRISM v. 4, and the % inhibition calculated from the top and bottom of the resulting curve.

## 4.3. Docking studies

Molecular docking studies were carried out to understand the molecular binding modes of the active synthesized compounds towards two different biological targets the human MAO enzymes. The Protein Data Bank (PDB) crystallographic structures of human MAO-A (PDB ID: 2BXR) and human MAO-B (PDB ID: 2BYB) were prepared by the removal of the co-crystallized ligands [42]. Before screening the new compounds, the docking protocol was validated by running the simulation using these ligands, and low RMSD between docked and crystal conformations was obtained. AutoDock 3.0 [43] and MOE [44] softwares were used for all docking calculations. The AutoDockTools package was employed to generate the docking input files and to analyse the docking results. A grid box size of $90 \times 90 \times 90$ points with a spacing of $0.375$ Å between the grid points was generated that covered almost the entire protein surface. All non-polar hydrogens and crystallographic water molecules were removed prior to the calculations. The docking grid was centred on the mass centre of the bound drugs. Ligands were fully flexibly docked. In each case, 100 docked structures were generated using genetic algorithm searches. A default protocol was applied with an initial population of 50 randomly placed conformations, a maximum number of $2.5 \times 10^5$ energy evaluations and a maximum number of $2.7 \times 10^4$ generations. Heavy atom comparison root mean square deviations (RMSD values) were calculated and initial ligand binding modes were plotted. Protein–ligand interaction plots were generated using MOE 2012 [45].

Data accessibility. Our spectral data are attached with the submission as suggested by the editor Dr Laura Smith.

Authors' contributions. A.A. planned the project, structural analysis and writing. A.H.H. and M.K.A. performed the synthetic organic part and writing. H.E.A.A. performed the docking analysis and writing. S.M.S. collected and analysed the X-ray data. A.M. did the MAO inhibition analysis. R.R.R. interpreted the biological results and wrote the manuscript. All authors gave final approval for publication.

Competing interests. There are no conflicts to declare.

Funding. Financial support came from the Deanship of Scientific Research at Taibah University, Al-Madinah Al-Munawarah, Saudi Arabia (project no. 7101). The funders did not influence the research or its reporting.

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
