## [Reviewer comments · Royal Society Open Science]

Review History

RSOS-191516.R0 (Original submission)

Review form: Reviewer 1

Is the manuscript scientifically sound in its present form?

No

Are the interpretations and conclusions justified by the results?

No

Is the language acceptable?

No

Do you have any ethical concerns with this paper?

No

Have you any concerns about statistical analyses in this paper?

No

Recommendation?

Reject

Comments to the Author(s)

In "Quinazoline analogues as ligands for monoamine oxidases: design, synthesis, and biological activity screening" by A. Amer, A. H. Hegazi, M. K. Alshekh, H. Ahmed, S. Soliman, A. Maniquet, and R. Ramsay submitted to Royal Society Open Science as a Research Article, the authors describe the synthesis, in vitro biological activity, and docking studies of a series of quinazoline derivatives as inhibitors for MAO-A and MAO-B with a focus on developing specific inhibitors for MAO-A. This manuscript, however, has many major issues (see major questions and comments below) which makes the analysis of the novelty and value of the work difficult. Thus, it is the opinion of this reviewer this manuscript in its current form should be rejected for publication in Royal Society Open Science.

Major questions and comments:

1. The title does not seem to describe the studies presented in the manuscript well. For example the focus is not to develop new substrates as the word ligand in the title would suggest but instead specific inhibitors for MAO-A over MAO-B.
2. Additionally, the design aspect of this manuscript is not discussed. Adding a clear design rationale towards the beginning of the manuscript would greatly strengthen this manuscript.
3. It would be nice if all figures, charts, etc had captions or more detailed captions.
4. In chart 1, the Y groups need to be defined.
5. On page 5, lines 2-12: when discussing design factors pointing out compounds by number (e.g., 5a) would make understanding what the authors are discussing easier on the reader.
6. Page 5 & 6, lines 36-36 and figure 1: The inclusion of the crystal structure of 4c as figure 1 with minimal discussion in text does not add to the manuscript especially in the absence of other crystal structures reported or absence discussion of any known crystal structures of previous compounds.
7. There is a figure inserted in page 6 that is not labeled as a figure and does not have a
8. Page 5-8, "Chemistry" section: this section overall reads as a rehash of the experimental section including details that are not important to a casual reader (one who is not seeking to replicate the work, these readers would be able to get this information in the experimental). Some examples include discussing very typical purification conditions and analysis techniques (page 7, lines 18-21) and how a compound was obtained and characterized from previously reported procedures (page 7, lines 36-46; page 8 lines 18-29). Overall this section could be compressed to a few paragraphs focusing on the discussion that does matter like the presence of E/Z isomers of 5a-5d.
9. Page 7, Scheme 2: condition details are missing such as solvent, time, temperature.
10. Page 8, Scheme 4: is missing conditions (and reagents) for each step of the reaction scheme shown.
11. Page 8, line 49: stating the actual concentrations used instead of $2 \times K_m$ tyramine would be more clear.
12. Page 9, line 7-10: "Incomplete inhibition of activity could arise either from indirect effects on these enzymes..." Is there experimental evidence or reference in the literature to support this statement?
13. Page 9, table 1: what is the error in the %inhibition experiment? How many times was this repeated?
14. Page 9, table 1, compound 6: why is a K_m value given in the IC50 column?
15. Page 10, line 54-55: "...gave a measurable rate of H₂O₂..." the details of this assay are not discussed in text, thus a reader with out experience in this particular assay would not be able to interpret what this means. Perhaps, discussing what the implications of a measurable rate of H₂O₂ mean would be more clear to a general reader.
16. Page 10 lines 54-55: and page 11 line 3. "Compound 6 (but not 12d) gave a measurable rate of H₂O₂ [...] as shown in Figure 2." Figure 2 does not show this data.
17. Page 11 lines 10-22: Does 3b also have some MAO-A inhibition activity?
18. Page 11, lines 24-26: "Considering first the phenylquinazoline head shown in the design..." Which compounds are specifically being referred to here?

19. Page 11, line 42-43: "This difference might be due to the changed interactions in the active site." This point is not discussed until later. Perhaps moving the SAR discussion until after figure 2 and docking results are discussed would be more appropriate and allow for the discussion of docking results in context of the SAR analysis.
20. Page 12, Figure 2: A key is needed: What do the different colors mean (green vs purple)? What do the different circle borders mean (blue and red)? What do double blue circles mean? What does the blue highlighting mean around certain bonds? What do the dotted lines mean?
21. Page 12, Figure 2 & Page 13: "Molecular docking simulation" While Figure 2 is nice to show general interactions to a general audience, detailed atomic level figures or the final 3D structure docking files should be provided in order to be able to appropriately determine if the analysis in this manuscript is valid.
22. Page 13, lines 6 to 18: Is there a reference that describes the binding pockets?
23. Page 13, lines 20-22: What are the hydrogen bond distances and exactly which atoms are the compounds hydrogen bonding to for Gln57 and Lys296 (e.g., backbone amide nitrogen, R-groups)?
24. Page 13, lines 25-26: how does the scaffold bind to the residues Gln206, Tyr326, Thr201, and Ile199?
25. Page 13, lines 26-27: how does the methyl group interact with Phe168 and Leu171?
26. Page 13, lines 27-29: how does 14b form unfavorable interactions in the pocket?
27. Page 13, lines 30-32: what are the hydrogen bonding distances and which atoms are involved in the interaction between 5g and Tyr69, FAD, and Gln74.
28. Page 13, lines 35-36: Which atoms are involved in the hydrogen bonding of Ser209.
29. Page 34, Scheme 3, specific conditions are needed for each step: reagents, time, temperature, solvents, etc.

Minor questions and comments:

1. Chart 1 is more like a figure than a chart
2. On page 5, line 11 "...which contain -HC=N- linker" using the accepted naming convention for this group would be more clear, much like the hydrazido functionality on line 5-6.
3. Page 5, line 16-17: the title of chemistry for this section is not typical convention, I would suggest the title of "Organic synthesis" for this section
4. Table 1 formatting is odd (maybe due to reformatting for upload?) with line breaks in middle of words, etc.
5. Page 9, table 1, footnote should be cited as a reference in the references section.
6. Page 13, lines 6-8: PDB IDs should be identified as a PDB ID (e.g., PDB: 2BXR)
7. Page 13, lines 6-8: proper citations should be provided for the PDB IDs
8. Page 13, lines 48-49: "Prior work showed.." needs a reference.
9. Page 28, lines 43: Is "...2.5X10⁵ energy..." supposed to read 2.5X10⁵ energy?
10. Page 28, lines 44: Is "...2.7X10⁴ generations" supposed to read 2.7X10⁴ generations?
11. Page 28, lines 48: MOE 2012 needs a reference.

Review form: Reviewer 2

Is the manuscript scientifically sound in its present form?

Yes

Are the interpretations and conclusions justified by the results?

Yes

Is the language acceptable?

Yes

Do you have any ethical concerns with this paper?

No

Have you any concerns about statistical analyses in this paper?

No

Recommendation?

Accept with minor revision (please list in comments)

Comments to the Author(s)

Based on previous studies by the authors group regarding a high-potency MAO-A inhibitor, the authors prepare and assay a series of new quinazoline-based inhibitors for MAO-A and MAO-B. The authors rationalize the observed potency among these compounds through a combination of molecular docking simulations and by comparative analysis among SAR variants.

The major concern I have is with regards to the purity of the derived quinazolines. From the reported synthesis (in the supporting information), analogs 5a-h and 12a-d are prepared as mixtures of E and Z isomers. Obviously, these structural isomers will have quite different affinities for the active site of MAO (A or B) and will also display different competencies as substrates for MAO oxidation. In the absence of isomerically pure compounds, the data becomes hard to validate especially as the ratio among E and Z isomers likely differs between structural variants based on the identity of the aryl component used (size and electronics of the aryl unit can dictate E-Z selectivity). If the samples are isomerically pure, than this information needs to be conveyed much more clearly by the authors. If not, the authors should consider reporting the ratio of E and Z isomers for each of the compounds surveyed in the main body of the manuscript and clearly label these within the spectrum in the supporting information section.

Of further note, the authors make comment of the "best inhibitor was 5d with sub-micromolar... and 3-fold selectivity for MAO-A". However, in Table 1, compound 5d inhibits hMAO-A and hMAO-B in a roughly 1:1 ratio. This is also the case when touting the selectivity for compound 5h. The authors may want to re-examine the numbering or compound listings, both in the main text and in the tables as the data provided does not seem to match the statements made in the main text.

To establish competitive inhibition (as suggested by the authors), the authors need to show their experimental results regarding the effect of reaction rate in the presence of increasing amounts of inhibitor and a constant amount of natural substrate (tyramine). Does the V_{max} remain constant while K_m increases? If V_{max} is decreasing as the authors state " V_{max} for the oxidation of 6 was 77% of the V_{max} for tyramine", wouldn't this imply the formation of an unproductive enzyme-inhibitor complex or enzyme deactivation? The decrease in V_{max} is more indicative of an uncompetitive or non-competitive inhibition mechanism and warrants further discussion.

As stated previously, due to concerns in isomeric purity between 5a-h and 12a-d, this makes accurate determination of IC_{50} values difficult. Unless these samples are isomerically pure, the authors should consider reporting K_i values for the various inhibitors to avoid substrate and inhibitor concentration errors.

Decision letter (RSOS-191516.R0)

19-Nov-2019

Dear Professor Amer:

Manuscript ID: RSOS-191516

Title: "Quinazoline analogues as ligands for monoamine oxidases: design, synthesis, and biological activity screening"

Thank you for submitting the above manuscript to Royal Society Open Science. Your paper was sent to reviewers and their comments are included at the bottom of this letter.

In view of the concerns raised by the reviewers, the manuscript has been rejected in its current form. However, a new manuscript may be submitted which takes into consideration these comments.

Please note that resubmitting your manuscript does not guarantee eventual acceptance, and that your resubmission will be subject to peer review before a decision is made.

Your resubmitted manuscript should be submitted by 18-May-2020. If you are unable to submit by this date please contact the Editorial Office.

On behalf of the Subject Editor Professor Anthony Stace and the Associate Editor Dr Andrew Harned

REVIEWER(S) REPORTS:

Associate Editor Comments to Author ():

RSC Associate Editor:

Comments to the Author:

The referees have raised a number of valid concerns. Some of these are more minor and relate to formatting, or how the data and results are presented. But, there are a number of more troubling concerns related to experimental design, compound characterization, and data interpretation. If the authors are able to address these concerns, I would be willing to send a substantially modified manuscript out for peer review. The authors should carefully consider and respond to all comments before submitting any modified paper.

RSC Subject Editor:

Comments to the Author:

(There are no comments.)

Reviewers' Comments to Author:

Reviewer: 1

Comments to the Author(s)

In "Quinazoline analogues as ligands for monoamine oxidases: design, synthesis, and biological activity screening" by A. Amer, A. H. Hegazi, M. K. Alshekh, H. Ahmed, S. Soliman, A. Maniquet, and R. Ramsay submitted to Royal Society Open Science as a Research Article, the authors describe the synthesis, in vitro biological activity, and docking studies of a series of quinazoline derivatives as inhibitors for MAO-A and MAO-B with a focus on developing specific inhibitors for MAO-A. This manuscript, however, has many major issues (see major questions and comments below) which makes the analysis of the novelty and value of the work difficult. Thus, it is the opinion of this reviewer this manuscript in its current form should be rejected for publication in Royal Society Open Science.

Major questions and comments:

1. The title does not seem to describe the studies presented in the manuscript well. For example the focus is not to develop new substrates as the word ligand in the title would suggest but instead specific inhibitors for MAO-A over MAO-B.
2. Additionally, the design aspect of this manuscript is not discussed. Adding a clear design rationale towards the beginning of the manuscript would greatly strengthen this manuscript.
3. It would be nice if all figures, charts, etc had captions or more detailed captions.
4. In chart 1, the Y groups need to be defined.
5. On page 5, lines 2-12: when discussing design factors pointing out compounds by number (e.g., 5a) would make understanding what the authors are discussing easier on the reader.
6. Page 5 & 6, lines 36-36 and figure 1: The inclusion of the crystal structure of 4c as figure 1 with minimal discussion in text does not add to the manuscript especially in the absence of other crystal structures reported or absence discussion of any known crystal structures of previous compounds.
7. There is a figure inserted in page 6 that is not labeled as a figure and does not have a
8. Page 5-8, "Chemistry" section: this section overall reads as a rehash of the experimental section including details that are not important to a casual reader (one who is not seeking to replicate the work, these readers would be able to get this information in the experimental). Some examples include discussing very typical purification conditions and analysis techniques (page 7, lines 18-21) and how a compound was obtained and characterized from previously reported procedures (page 7, lines 36-46; page 8 lines 18-29). Overall this section could be compressed to a few paragraphs focusing on the discussion that does matter like the presence of E/Z isomers of 5a-5d.
9. Page 7, Scheme 2: condition details are missing such as solvent, time, temperature.
10. Page 8, Scheme 4: is missing conditions (and reagents) for each step of the reaction scheme shown.
11. Page 8, line 49: stating the actual concentrations used instead of 2xKm tyramine would be more clear.
12. Page 9, line 7-10: "Incomplete inhibition of activity could arise either from indirect effects on these enzymes..." Is there experimental evidence or reference in the literature to support this statement?
13. Page 9, table 1: what is the error in the %inhibition experiment? How many times was this repeated?
14. Page 9, table 1, compound 6: why is a Km value given in the IC50 column?
15. Page 10, line 54-55: "...gave a measurable rate of H2O2..." the details of this assay are not discussed in text, thus a reader with out experience in this particular assay would not be able to interpret what this means. Perhaps, discussing what the implications of a measurable rate of H2O2 mean would be more clear to a general reader.
16. Page 10 lines 54-55: and page 11 line 3. "Compound 6 (but not 12d) gave a measurable rate of H2O2 [...] as shown in Figure 2." Figure 2 does not show this data.
17. Page 11 lines 10-22: Does 3b also have some MAO-A inhibition activity?

18. Page 11, lines 24-26: "Considering first the phenylquinazoline head shown in the design..." Which compounds are specifically being referred to here?
19. Page 11, line 42-43: "This difference might be due to the changed interactions in the active site." This point is not discussed until later. Perhaps moving the SAR discussion until after figure 2 and docking results are discussed would be more appropriate and allow for the discussion of docking results in context of the SAR analysis.
20. Page 12, Figure 2: A key is needed: What do the different colors mean (green vs purple)? What do the different circle borders mean (blue and red)? What do double blue circles mean? What does the blue highlighting mean around certain bonds? What do the dotted lines mean?
21. Page 12, Figure 2 & Page 13: "Molecular docking simulation" While Figure 2 is nice to show general interactions to a general audience, detailed atomic level figures or the final 3D structure docking files should be provided in order to be able to appropriately determine if the analysis in this manuscript is valid.
22. Page 13, lines 6 to 18: Is there a reference that describes the binding pockets?
23. Page 13, lines 20-22: What are the hydrogen bond distances and exactly which atoms are the compounds hydrogen bonding to for Gln57 and Lys296 (e.g., backbone amide nitrogen, R-groups)?
24. Page 13, lines 25-26: how does the scaffold bind to the residues Gln206, Tyr326, Thr201, and Ile199?
25. Page 13, lines 26-27: how does the methyl group interact with Phe168 and Leu171?
26. Page 13, lines 27-29: how does 14b form unfavorable interactions in the pocket?
27. Page 13, lines 30-32: what are the hydrogen bonding distances and which atoms are involved in the interaction between 5g and Tyr69, FAD, and Gln74.
28. Page 13, lines 35-36: Which atoms are involved in the hydrogen bonding of Ser209.
29. Page 34, Scheme 3, specific conditions are needed for each step: reagents, time, temperature, solvents, etc.

Minor questions and comments:

1. Chart 1 is more like a figure than a chart
2. On page 5, line 11 "...which contain -HC=N- linker" using the accepted naming convention for this group would be more clear, much like the hydrazido functionality on line 5-6.
3. Page 5, line 16-17: the title of chemistry for this section is not typical convention, I would suggest the title of "Organic synthesis" for this section
4. Table 1 formatting is odd (maybe due to reformatting for upload?) with line breaks in middle of words, etc.
5. Page 9, table 1, footnote should be cited as a reference in the references section.
6. Page 13, lines 6-8: PDB IDs should be identified as a PDB ID (e.g., PDB: 2BXR)
7. Page 13, lines 6-8: proper citations should be provided for the PDB IDs
8. Page 13, lines 48-49: "Prior work showed..." needs a reference.
9. Page 28, lines 43: Is "...2.5X10⁵ energy..." supposed to read 2.5X10⁵ energy?
10. Page 28, lines 44: Is "...2.7X10⁴ generations" supposed to read 2.7X10⁴ generations?
11. Page 28, lines 48: MOE 2012 needs a reference.

Reviewer: 2

Comments to the Author(s)

Based on previous studies by the authors group regarding a high-potency MAO-A inhibitor, the authors prepare and assay a series of new quinazoline-based inhibitors for MAO-A and MAO-B. The authors rationalize the observed potency among these compounds through a combination of molecular docking simulations and by comparative analysis among SAR variants.

The major concern I have is with regards to the purity of the derived quinazolines. From the reported synthesis (in the supporting information), analogs 5a-h and 12a-d are prepared as mixtures of E and Z isomers. Obviously, these structural isomers will have quite different affinities for the active site of MAO (A or B) and will also display different competencies as

substrates for MAO oxidation. In the absence of isomerically pure compounds, the data becomes hard to validate especially as the ratio among E and Z isomers likely differs between structural variants based on the identity of the aryl component used (size and electronics of the aryl unit can dictate E-Z selectivity). If the samples are isomerically pure, than this information needs to be conveyed much more clearly by the authors. If not, the authors should consider reporting the ratio of E and Z isomers for each of the compounds surveyed in the main body of the manuscript and clearly label these within the spectrum in the supporting information section.

Of further note, the authors make comment of the "best inhibitor was 5d with sub-micromolar... and 3-fold selectivity for MAO-A". However, in Table 1, compound 5d inhibits hMAO-A and hMAO-B in a roughly 1:1 ratio. This is also the case when touting the selectivity for compound 5h. The authors may want to re-examine the numbering or compound listings, both in the main text and in the tables as the data provided does not seem to match the statements made in the main text.

To establish competitive inhibition (as suggested by the authors), the authors need to show their experimental results regarding the effect of reaction rate in the presence of increasing amounts of inhibitor and a constant amount of natural substrate (tyramine). Does the V_{max} remain constant while K_m increases? If V_{max} is decreasing as the authors state " V_{max} for the oxidation of 6 was 77% of the V_{max} for tyramine", wouldn't this imply the formation of an unproductive enzyme-inhibitor complex or enzyme deactivation? The decrease in V_{max} is more indicative of an uncompetitive or non-competitive inhibition mechanism and warrants further discussion.

As stated previously, due to concerns in isomeric purity between 5a-h and 12a-d, this makes accurate determination of IC_{50} values difficult. Unless these samples are isomerically pure, the authors should consider reporting K_i values for the various inhibitors to avoid substrate and inhibitor concentration errors.

Author's Response to Decision Letter for (RSOS-191516.R0)

See Appendix A.

RSOS-200050.R0

Review form: Reviewer 1

Is the manuscript scientifically sound in its present form?

No

Are the interpretations and conclusions justified by the results?

Yes

Is the language acceptable?

Yes

Do you have any ethical concerns with this paper?

No

Have you any concerns about statistical analyses in this paper?

No

Recommendation?

Major revision is needed (please make suggestions in comments)

Comments to the Author(s)

In the now titled "Design, synthesis, molecular modeling, and in vitro screening of monoamine oxidase inhibitory activities of novel quinazolyyl hydrazine derivatives" by A. Amer, A. H. Hegazi, M. K. Alshekh, H. E. A. Ahmed, S. M. Soliman, A. Maniquet, and R. Ramsay submitted to Royal Society Open Science as a Research Article, the authors describe the synthesis, in vitro biological activity, and docking studies of a series of quinazoline derivatives as inhibitors for MAO-A and MAO-B with a focus on developing specific inhibitors for MAO-A. The revised manuscript greatly improves on their previous submission, however, concerns about isomeric purity, the role of each isomer in the inhibitory activity of the compounds, and how the different %ee may affect comparisons remains (See Major Questions and Comments #3). The authors have addressed all of my other concerns except as detailed below. Thus, I recommend that this manuscript be accepted after major revisions with the inclusion of an experiment that addresses this issue.

Major Questions and Comments:

1. Response to Reviewer #1 comments, major questions and comments #22/ page 10, lines 5-16: A description of the binding pocket for MAO-A is still missing despite being marked done. An explanation of MAO-B is present, it would be nice to have a similar description of MAO-A. This information is important in understanding how the new inhibitors interact with MAO-A.
2. I really appreciate the work to place a detailed analysis of the docking studies in Table S2. This makes it much easier to understand the results of the docking studies for an interested reader.
3. Response to Reviewer #2 comments, Reviewer 2 makes a good point in pointing out concerns with isomeric purity. Could the difference in activity between each compound depend on isomeric purity? Including the context of enantiomeric purity when discussing comparisons of inhibitory activity would be nice. Additionally, if possible, the assay could be ran with an enantiomerically pure compound as an example (or significantly higher enantiomeric excess) to demonstrate if the enantiomeric purity matters or not would be best to address these concerns. In absence of this, perhaps docking studies could suggest which isomer is the active one.

Minor Questions and Comments:

1. Response to Reviewer #1 comments, minor questions and comments #1, I'll leave the determination of if Chart 1 is appropriately labeled as a chart with the editor, but still believe that it is more suitable as a figure (though this doesn't really matter at all for suitability for publication).
2. Page 12, Figure 4, dG should be written with the delta symbol, additionally units for the deltaGs should be included (e.g., kJ/mol, kcal/mol).
3. Response to Reviewer #1 comments, minor questions and comments #10, it is unclear if the 2.7¹⁰⁴ is a typo or correct from the author's response, the manuscript does not reflect a change in the value.

Review form: Reviewer 2**Is the manuscript scientifically sound in its present form?**

Yes

Are the interpretations and conclusions justified by the results?

Yes

Is the language acceptable?

Yes

Do you have any ethical concerns with this paper?

No

Have you any concerns about statistical analyses in this paper?

No

Recommendation?

Accept as is

Comments to the Author(s)

The authors clearly addressed my previous concerns and I believe the interpretation of the data is well reasoned and accurate. I wish to thank the authors for their time in addressing my concerns and for their contribution to the scientific literature.

Decision letter (RSOS-200050.R0)

11-Feb-2020

Dear Professor Amer:

Title: Quinazoline analogues as ligands for monoamine oxidases: design, synthesis, and biological activity screening

Manuscript ID: RSOS-200050

Thank you for submitting the above manuscript to Royal Society Open Science. On behalf of the Editors and the Royal Society of Chemistry, I am pleased to inform you that your manuscript will be accepted for publication in Royal Society Open Science subject to minor revision in accordance with the referee suggestions. Please find the reviewers' comments at the end of this email.

The reviewers and handling editors have recommended publication, but also suggest some minor revisions to your manuscript. Therefore, I invite you to respond to the comments and revise your manuscript.

Because the schedule for publication is very tight, it is a condition of publication that you submit the revised version of your manuscript before 20-Feb-2020. Please note that the revision deadline will expire at 00.00am on this date. If you do not think you will be able to meet this date please let me know immediately.

Best wishes,
Dr Laura Smith
Publishing Editor, Journals

On behalf of the Subject Editor Professor Anthony Stace and the Associate Editor Dr Andrew Harned.

RSC Associate Editor

Comments to the Author:

The authors have submitted a revised manuscript based on the comments raised during the previous review. Reviewer 2 has stated that they are satisfied with the authors' response. Reviewer 1 has some additional questions/concerns. I have commented on these below:

- (1) Adding a similar description of MAO-A to the first paragraph of Section 2.3 would be helpful for understanding the differences between MAO-A and MAO-B. This would be particularly helpful for non-specialist readers.
- (2) Reviewer 1 again raises concerns regarding isomeric purity. I am inclined to defer to Reviewer

2's satisfaction on this issue as this is a somewhat preliminary report and the activities as a whole are only in the micromolar range. Also the difference in the E/Z ratios are rather minor. Their comments regarding enantiomeric purity are curious as all of the chiral compounds are racemic. Again, having a better understanding of the activity of the different enantiomers will be important for further studies, but at this point I do not feel it is needed.

(3) I am fine with Chart 1 being labeled as it is.

(4) I agree that the delta symbol should be used where appropriate in Figure 4. The units should be listed as well. Also, the graphics as a whole for this figure are of low quality. Please submit a graphic with better resolution and is more readable.

(5) Clarify Reviewer 1's concern raised in the last question.

Reviewer comments to Author:

Reviewer: 1

Comments to the Author(s)

In the now titled "Design, synthesis, molecular modeling, and in vitro screening of monoamine oxidase inhibitory activities of novel quinazolyl hydrazine derivatives" by A. Amer, A. H. Hegazi, M. K. Alshekh, H. E. A. Ahmed, S. M. Soliman, A. Maniquet, and R. Ramsay submitted to Royal Society Open Science as a Research Article, the authors describe the synthesis, in vitro biological activity, and docking studies of a series of quinazoline derivatives as inhibitors for MAO-A and MAO-B with a focus on developing specific inhibitors for MAO-A. The revised manuscript greatly improves on their previous submission, however, concerns about isomeric purity, the role of each isomer in the inhibitory activity of the compounds, and how the different %ee may affect comparisons remains (See Major Questions and Comments #3). The authors have addressed all of my other concerns except as detailed below. Thus, I recommend that this manuscript be accepted after major revisions with the inclusion of an experiment that addresses this issue.

Major Questions and Comments:

1. Response to Reviewer #1 comments, major questions and comments #22/ page 10, lines 5-16: A description of the binding pocket for MAO-A is still missing despite being marked done. An explanation of MAO-B is present, it would be nice to have a similar description of MAO-A. This information is important in understanding how the new inhibitors interact with MAO-A.
2. I really appreciate the work to place a detailed analysis of the docking studies in Table S2. This makes it much easier to understand the results of the docking studies for an interested reader.
3. Response to Reviewer #2 comments, Reviewer 2 makes a good point in pointing out concerns with isomeric purity. Could the difference in activity between each compound depend on isomeric purity? Including the context of enantiomeric purity when discussing comparisons of inhibitory activity would be nice. Additionally, if possible, the assay could be ran with an enantiomerically pure compound as an example (or significantly higher enantiomeric excess) to demonstrate if the enantiomeric purity matters or not would be best to address these concerns. In absence of this, perhaps docking studies could suggest which isomer is the active one.

Minor Questions and Comments:

1. Response to Reviewer #1 comments, minor questions and comments #1, I'll leave the determination of if Chart 1 is appropriately labeled as a chart with the editor, but still believe that it is more suitable as a figure (though this doesn't really matter at all for suitability for publication).
2. Page 12, Figure 4, dG should be written with the delta symbol, additionally units for the deltaGs should be included (e.g., kJ/mol, kcal/mol).

3. Response to Reviewer #1 comments, minor questions and comments #10, it is unclear if the 2.7^{104} is a typo or correct from the author's response, the manuscript does not reflect a change in the value.

Reviewer: 2

Comments to the Author(s)

The authors clearly addressed my previous concerns and I believe the interpretation of the data is well reasoned and accurate. I wish to thank the authors for their time in addressing my concerns and for their contribution to the scientific literature.

Author's Response to Decision Letter for (RSOS-200050.R0)

See Appendix B.

Decision letter (RSOS-200050.R1)

28-Feb-2020

Dear Professor Amer:

Title: Design, synthesis, molecular modelling and in vitro screening of monoamine oxidase inhibitory activities of novel quinazolyl hydrazine derivatives

Manuscript ID: RSOS-200050.R1

It is a pleasure to accept your manuscript in its current form for publication in Royal Society Open Science. The chemistry content of Royal Society Open Science is published in collaboration with the Royal Society of Chemistry.

On behalf of the Subject Editor Professor Anthony Stace and the Associate Editor Dr Andrew Harned.

RSC Associate Editor

Comments to the Author:

The authors appear to have addressed the concerns raised in the previous review. I believe the manuscript to be acceptable for publication in its current form.

Reviewer(s)' Comments to Author:

Appendix A

Responses to Reviewer # 1

1-The title does not seem to describe the studies presented in the manuscript well. For example, the focus is not to develop new substrates as the word ligand in the title would suggest but instead specific inhibitors for MAO-A over MAO-B.

The title of the manuscript has been changed to “Design, Synthesis, Molecular Modeling and *in vitro* Screening of Monoamine Oxidase Inhibitory Activities of Novel Quinazolyhydrazine Derivatives”

2- Additionally, the design aspect of this manuscript is not discussed. Adding a clear design rationale towards the beginning of the manuscript would greatly strengthen this manuscript.

Done

3- It would be nice if all figures, charts, etc had captions or more detailed captions.

Done

4-In chart 1, the Y groups need to be defined

Done

5- On page 5, lines 2-12: when discussing design factors pointing out compounds by number (e.g., 5a) would make understanding what the authors are discussing easier on the reader.

Done

6- Page 5 & 6, lines 36-36 and figure 1: The inclusion of the crystal structure of 4c as figure 1 with minimal discussion in text does not add to the manuscript especially in the absence of other crystal structures reported or absence discussion of any known crystal structures of previous compounds.

X-ray crystallography was meant to secure the structure and data was presented in the supplementary materials

7- There is a figure inserted in page 6 that is not labeled as a figure and does not have a

Done [Fig. 2]

8- Page 5-8, “Chemistry” section: this section overall reads as a rehash of the experimental section including details that are not important to a casual reader (one who is not seeking to replicate the work, these readers would be able to get this information in the experimental). Some examples include discussing very typical purification conditions and analysis techniques (page 7, lines 18-21) and how a compound was obtained and characterized from previously reported procedures (page 7, lines 36-46; page 8 lines 18-

29). Overall this section could be compressed to a few paragraphs focusing on the discussion that does matter like the presence of E/Z isomers of 5a-5d.

Done

9- Page 7, Scheme 2: condition details are missing such as solvent, time, temperature.

Done

10- Page 8, Scheme 4: is missing conditions (and reagents) for each step of the reaction scheme shown.

Done

11- Page 8, line 49: stating the actual concentrations used instead of 2xKm tyramine would be more clear.

Actual concentrations used (already present in Methods) have been added to the main text. The “2xKm” has been retained because this is definitive for optimum screening of reversible inhibitors.

Done

12- Page 9, line 7-10: “Incomplete inhibition of activity could arise either from indirect effects on these enzymes...” Is there experimental evidence or reference in the literature to support this statement?

There is a wealth of information on the complexities of enzyme kinetics in the literature. Two articles that mention incomplete or active vs non-active site interactions are:

McDonald, A.; Tipton, K., Kinetics of Catalyzed Reactions – Biological. In Encyclopedia of Catalysis, Horvath, I. T., Ed. John Wiley & Sons, Inc: 2002, doi: 10.1002/0471227617.eoc127

Blat, Y. Chem Biol Drug Des 2010; 75: 535–540, doi: 10.1111/j.1747-0285.2010.00972.x

These are generally accepted suggestions so no reference has been added to the manuscript.

Done

13- Page 9, table 1: what is the error in the % inhibition experiment? How many times was this repeated? Average of two experiments (each with 2 technical replicates) with <15% difference (range 1-15%). A statement has been added to the table footnote.

Done

14- Page 9, table 1, compound 6: why is a Km value given in the IC₅₀ column?

Compound 6 is a substrate that increases the observed rate rather than decreasing it. Compound 6 was assessed as a substrate (no tyramine present) to obtain the Km (and

V_{max}). For $E + S \rightleftharpoons ES \rightarrow EP$, $K_m = (k_{-1} + k_2)/k_1$, where k_1 is the on rate, k_{-1} the off rate and k_2 the rate of the chemical reaction. Thus assuming that k_2 is slow compared to the on and off rates, K_m is a parameter similar to K_i and so is provided for comparison with the inhibitory binding of the other compounds.

The K_m value is now given as a footnote and further explanation added to the text (see next comment). Figure 2 shows the determination of the K_m .

Done

15- Page 10, line 54-55: “..gave a measurable rate of H₂O₂...” the details of this assay are not discussed in text, thus a reader without experience in this particular assay would not be able to interpret what this means. Perhaps, discussing what the implications of a measurable rate of H₂O₂ mean would be more clear to a general reader.

A fuller description has been added: ” Compound **6** in the screening assay with 10 μ M inhibitor gave an increased rate of appearance of fluorescence indicating faster production of the product H₂O₂ (not the slower rate observed with other compounds that inhibit MAO activity). When incubated with MAO-A in the absence of the normal substrate, **6** (but not **12d**) gave a measurable rate of H₂O₂ production, as shown in Figure 2. Thus, **6** binds to and is oxidized by MAO-A.”

Done

16- Page 10 lines 54-55: and page 11 line 3. “Compound **6** (but not **12d**) gave a measurable rate of H₂O₂ [...] as shown in Figure 2.” Figure 2 does not show this data.

The intended Figure 2 has been inserted.

Done

17- Page 11 lines 10-22: Does **3b** also have some MAO-A inhibition activity?

Compound **3b** gave only 10% inhibition of MAO A activity at 10 μ M (see Table 1) so was not further investigated.

18- Page 11, lines 24-26: “Considering first the phenylquinazoline head shown in the design...” Which compounds are specifically being referred to here?

Type **5** compounds (X= H, Cl). “Type 5 compounds” has been added to the text.

Done

19- Page 11, line 42-43: “This difference might be due to the changed interactions in the active site.” This point is not discussed until later. Perhaps moving the SAR discussion until after figure 2 and docking results are discussed would be more appropriate and allow for the discussion of docking results in context of the SAR analysis.

The SAR discussion has been moved to after the docking analysis.

Done

20. Page 12, Figure 2: A key is needed: What do the different colors mean (green vs purple)? What do the different circle borders mean (blue and red)? What do double blue circles mean? What does the blue highlighting mean around certain bonds? What do the dotted lines mean?

Done

21. Page 12, Figure 2 & Page 13: “Molecular docking simulation” While Figure 2 is nice to show general interactions to a general audience, detailed atomic level figures or the final 3D structure docking files should be provided in order to be able to appropriately determine if the analysis in this manuscript is valid.

Done (Table S2)

22. Page 13, lines 6 to 18: Is there a reference that describes the binding pockets?

Done (Table S2)

23. Page 13, lines 20-22: What are the hydrogen bond distances and exactly which atoms are the compounds hydrogen bonding to for Gln57 and Lys296 (e.g., backbone amide nitrogen, R-groups)?

Done (Table S2)

24. Page 13, lines 25-26: how does the scaffold bind to the residues Gln206, Tyr326, Thr201, and Ile199?

Done (Table S2)

25. Page 13, lines 26-27: how does the methyl group interact with Phe168 and Leu171?

Done (Table S2)

26. Page 13, lines 27-29: how does 14b form unfavorable interactions in the pocket?

Done (Table S2)

27. Page 13, lines 30-32: what are the hydrogen bonding distances and which atoms are involved in the interaction between 5g and Tyr69, FAD, and Gln74.

Done (Table S2)

28. Page 13, lines 35-36: Which atoms are involved in the hydrogen bonding of Ser209.

Done (Table S2)

29. Page 34, Scheme 3, specific conditions are needed for each step: reagents, time, temperature, solvents, etc.

Done

Minor questions and comments:

1. Chart 1 is more like a figure than a chart

It is fine as a chart

2. On page 5, line 11 "...which contain -HC=N- linker" using the accepted naming convention for this group would be more clear, much like the hydrazido functionality on line 5-6.

Done

3. Page 5, line 16-17: the title of chemistry for this section is not typical convention, I would suggest the title of "Organic synthesis" for this section

Done

4. Table 1 formatting is odd (maybe due to reformatting for upload?) with line breaks in middle of words, etc.

Reformatted

Done

5. Page 9, table 1, footnote should be cited as a reference in the references section.

Done

6. Page 13, lines 6-8: PDB IDs should be identified as a PDB ID (e.g., PDB: 2BXR)

Done

7. Page 13, lines 6-8: proper citations should be provided for the PDB IDs

Done (REF 41)

8. Page 13, lines 48-49: "Prior work showed.." needs a reference.

Done (REF 24 COVERS)

9. Page 28, lines 43: Is "...2.5X10⁵ energy..." supposed to read 2.5X10⁵ energy?

Done

10. Page 28, lines 44: Is "...2.7X10⁴ generations" supposed to read 2.7X10⁴ generations?

Done

11. Page 28, lines 48: MOE 2012 needs a reference.

Done (REF 43)

Responses to Reviewer # 2

Comments to the Author(s)

Based on previous studies by the authors group regarding a high-potency MAO-A inhibitor, the authors prepare and assay a series of new quinazoline-based inhibitors for MAO-A and MAO-B. The authors rationalize the observed potency among these compounds through a combination of molecular docking simulations and by comparative analysis among SAR variants.

The major concern I have is with regards to the purity of the derived quinazolines. From the reported synthesis (in the supporting information), analogs 5a-h and 12a-d are prepared as mixtures of E and Z isomers. Obviously, these structural isomers will have quite different affinities for the active site of MAO (A or B) and will also display different competencies as substrates for MAO oxidation. In the absence of isomerically pure compounds, the data becomes hard to validate especially as the ratio among E and Z isomers likely differs between structural variants based on the identity of the aryl component used (size and electronics of the aryl unit can dictate E-Z selectivity). If the samples are isomerically pure, than this information needs to be conveyed much more clearly by the authors. If not, the authors should consider reporting the ratio of E and Z isomers for each of the compounds surveyed in the main body of the manuscript and clearly label these within the spectrum in the supporting information section.

The E/Z ratio was presented in Schemes 1 and 3

Of further note, the authors make comment of the "best inhibitor was 5d with sub-micromolar... and 3-fold selectivity for MAO-A". However, in Table 1, compound 5d inhibits hMAO-A and hMAO-B in a roughly 1:1 ratio. This is also the case when touting the selectivity for compound 5h. The authors may want to re-examine the numbering or compound listings, both in the main text and in the tables as the data provided does not seem to match the statements made in the main text.

It is true that 5d and 5h give the same (perhaps maximal?), % inhibition of MAO-A and MAO-B the screening assay using a single inhibitor concentration. This experiment was designed to show inhibition or no inhibition but the numbers are included because all compounds were tested. However, only those which gave some inhibition were further assessed. The more precise evaluation of IC₅₀ show that 5h inhibition remains roughly equal but 5d is more effective on MAO-A. In the IC₅₀ experiments, the bottom of the sigmoidal dose-response curve was around 90% for these inhibitors.

To establish competitive inhibition (as suggested by the authors), the authors need to show their experimental results regarding the effect of reaction rate in the presence of increasing amounts of inhibitor and a constant amount of natural substrate (tyramine). Does the V_{max} remain constant while K_m increases?

A full matrix varying both inhibitor and substrate was not done. This statement has been changed to “suggesting competitive inhibition” because increasing rate with increasing substrate is a valid quick test for inhibitor displacement.

If V_{max} is decreasing as the authors state "V_{max} for the oxidation of 6 was 77% of the V_{max} for tyramine", wouldn't this imply the formation of an unproductive enzyme-inhibitor complex or enzyme deactivation? The decrease in V_{max} is more indicative of an uncompetitive or non-competitive inhibition mechanism and warrants further discussion.

This is NOT inhibition – it is a different situation in that zero tyramine is present in the assay with **6** varied and zero **6** is present in the assay with tyramine varied. **Compound 6 is a substrate for MAO-A**. The experiment shown in Figure 2 is a typical Michaelis-Menten curve for MAO-A oxidizing **6**. The assay measures relative fluorescence units, so the comparison was made with tyramine run in parallel. That saturating concentration of **6** gives a rate 77% of the rate achieved with saturating concentration of tyramine means that it is a good substrate.

As stated previously, due to concerns in isomeric purity between 5a-h and 12a-d, this makes accurate determination of IC₅₀ values difficult. Unless these samples are isomerically pure, the authors should consider reporting K_i values for the various inhibitors to avoid substrate and inhibitor concentration errors.

The reviewer should note that IC₅₀ values are always dependent on the experimental conditions including substrate concentrations. It reports the inhibition by the mixture (about 70/30 for series 5 and 80/20 for series 12). Inhibitor concentrations are varied to determine IC₅₀. K_i values are rarely used in screening work because of the added material and time consumed to obtain them. The use of Cheng-Prusoff calculation to give K_i is only valid if simple Michaelis-Menten conditions are fulfilled. This is not true for MAO-B where clear evidence is present that inhibitors can bind to both MAO-B (oxidized) and MAO-B (reduced). Thus, when comparing MAO-A and MAO-B in screening there is no

advantage in accuracy to convert to K_i values. The experiments were conducted under identical conditions so any errors are equal for both MAO A and MAO B.

Done

Appendix B

Responses to RSC Associate Editor

(1) Adding a similar description of MAO-A to the first paragraph of Section 2.3 would be helpful for understanding the differences between MAO-A and MAO-B. This would be particularly helpful for non-specialist readers.

One sentence about MAO-A was added (and the reference reiterated) on page 10 in the section relating to the molecular docking simulation.

(2) Reviewer 1 again raises concerns regarding isomeric purity. I am inclined to defer to Reviewer 2's satisfaction on this issue as this is a somewhat preliminary report and the activities as a whole are only in the micromolar range. Also the difference in the E/Z ratios are rather minor. Their comments regarding enantiomeric purity are curious as all of the chiral compounds are racemic. Again, having a better understanding of the activity of the different enantiomers will be important for further studies, but at this point I do not feel it is needed.

Accepted by the Editor

(3) I am fine with Chart 1 being labeled as it is.

Accepted by the Editor

(4) I agree that the delta symbol should be used where appropriate in Figure 4. The units should be listed as well. Also, the graphics as a whole for this figure are of low quality. Please submit a graphic with better resolution and is more readable.

The figure was modified to high quality one and symbols were inserted.

(5) Clarify Reviewer 1's concern raised in the last question.

It is clear from the docking results in figure 4 that the most active binding pose of compound **14** is connected to the configuration *E* (**added**) while in case of type **5** compounds are the *z* isomers (**added**).

Responses to Reviewer # 1

Major Questions and Comments:

1. Response to Reviewer #1 comments, major questions and comments #22/ page 10, lines 5-16: A description of the binding pocket for MAO-A is still missing despite being marked done. An explanation of MAO-B is present, it would be nice to have a similar description of MAO-A. This information is important in understanding how the new inhibitors interact with MAO-A.

It was done by adding MAO-A description in the manuscript.

2. I really appreciate the work to place a detailed analysis of the docking studies in Table S2. This makes it much easier to understand the results of the docking studies for an interested reader.

The table S2 was moved inside the manuscript as Table 2 to give detailed analyses of docking studies.

3. Response to Reviewer #2 comments, Reviewer 2 makes a good point in pointing out concerns with isomeric purity. Could the difference in activity between each compound depend on isomeric purity? Including the context of enantiomeric purity when discussing comparisons of inhibitory activity would be nice. Additionally, if possible, the assay could be ran with an enantiomerically pure compound as an example (or significantly higher enantiomeric excess) to demonstrate if the enantiomeric purity matters or not would be best to address these concerns. **In absence of this, perhaps docking studies could suggest which isomer is the active one.**

It is clear from the docking results in figure 4 that the most active binding pose of compound **14** is connected to the configuration *E* while in case of type **5** compounds are the *z* isomers.

Minor Questions and Comments:

1. Response to Reviewer #1 comments, minor questions and comments #1, I'll leave the determination of if Chart 1 is appropriately labeled as a chart with the editor, but still believe that it is more suitable as a figure (though this doesn't really matter at all for suitability for publication).

Accepted by the Editor

2. Page 12, Figure 4, ΔG should be written with the delta symbol, additionally units for the ΔG s should be included (e.g., kJ/mol, kcal/mol).

The symbol was added to Fig 4 and also units, Kcal/mol.

3. Response to Reviewer #1 comments, minor questions and comments #10, it is unclear if the 2.7^{104} is a typo or correct from the author's response, the manuscript does not reflect a change in the value.

Corrected. This is the energy evaluations for the docking run with indicated maximum generations of conformers, this is a default adjusted. (Corrected)